# Damage-induced basal epithelial cell migration modulates the spatial organization of redox signaling and sensory neuron regeneration

**Alexandra M Fister[1,2†], Adam Horn[1†], Michael R Lasarev[3], Anna Huttenlocher[1,4*]**

[1]Department of Medical Microbiology and Immunology, University of Wisconsin-Madison, Madison, United States; [2]Cellular and Molecular Biology Graduate Program, University of Wisconsin-Madison, Madison, United States; [3]Department of Biostatistics and Medical Informatics, University of Wisconsin-Madison, Madison, United States; [4]Department of Pediatrics, University of Wisconsin-Madison, Madison, United States

**\*For correspondence:**
huttenlocher@wisc.edu

[†]These authors contributed equally to this work

**Competing interest:** The authors declare that no competing interests exist.

**Abstract** Epithelial damage leads to early reactive oxygen species (ROS) signaling, which regulates sensory neuron regeneration and tissue repair. How the initial type of tissue injury influences early damage signaling and regenerative growth of sensory axons remains unclear. Previously we reported that thermal injury triggers distinct early tissue responses in larval zebrafish. Here, we found that thermal but not mechanical injury impairs sensory axon regeneration and function. Real-time imaging revealed an immediate tissue response to thermal injury characterized by the rapid Arp2/3-dependent migration of keratinocytes, which was associated with tissue scale ROS production and sustained sensory axon damage. Isotonic treatment was sufficient to limit keratinocyte movement, spatially restrict ROS production, and rescue sensory neuron function. These results suggest that early keratinocyte dynamics regulate the spatial and temporal pattern of long-term signaling in the wound microenvironment during tissue repair.

## eLife assessment

This **important** study identifies a novel link between the early keratinocyte response to wounds and the subsequent regenerative capacity of local sensory neurons. The evidence supporting the claims of the authors is **convincing**, although inclusion of conditional genetics or cell-autonomy tests would have strengthened the mechanistic aspects. The work will be of interest to cell and developmental biologists interested in tissue regeneration and cell interactions in a broader context.

## Introduction

Restoration of tissue function following epithelial injury requires the regeneration and activity of peripheral sensory neurons, which innervate the skin (*Mullen et al., 1996*; *Rabiller et al., 2021*; *Simões et al., 2014*). In response to tissue damage, axonal regeneration requires the clearance of axon fragments by phagocytes followed by new axonal growth (*Rasmussen et al., 2015*). While peripheral sensory neurons maintain a cell-intrinsic ability to regenerate following injury (*Liu et al., 2011*; *Mahar and Cavalli, 2018*; *Renthal et al., 2020*), local environmental cues also regulate the response of sensory neurons to tissue damage in vivo. Recent advances have identified the contribution of supporting cell populations, paracrine biochemical signaling, and biophysical interactions in

regulating peripheral sensory axon regeneration (*Avraham et al., 2021*; *Cheah et al., 2017*; *Villegas et al., 2012*). Despite these advances, we have limited understanding of how early collective signaling in the wound microenvironment is organized and leads to sensory axon regeneration and tissue repair.

Keratinocytes are a primary constituent of epithelial tissue and play a critical role in wound healing. In addition to mediating wound closure by actively migrating toward the site of injury, keratinocytes also generate pro-reparative signals such as reactive oxygen species (ROS), which coordinate longer-term repair pathways (*Dunnill et al., 2017*; *Linley et al., 2012*; *Love et al., 2013*; *Mittal et al., 2014*; *Romero et al., 2018*; *Sies and Jones, 2020*). While transient and localized ROS production promotes regeneration and sensory axon regrowth, chronically elevated ROS is associated with neurodegeneration and disease (*Cobb and Cole, 2015*; *Rieger and Sagasti, 2011*). Thus, precise temporal and spatial organization of tissue redox signaling is likely critical for efficient sensory neuron regeneration and tissue repair.

While epithelial tissue is well adapted to repair from mechanical damage, burn wounds heal poorly. Thermal injury results in chronic pain and lack of sensation in the affected tissue, suggesting that an abnormal sensory neuron response contributes to burn wound pathophysiology (*Blais et al., 2013*; *Choinière et al., 1991*; *Pavoni et al., 2010*; *Tirado-Esteban et al., 2020*; *Tsolakidis et al., 2022*). Despite this, we lack an understanding of why sensory neuron function is impaired after burn. Our previous work has demonstrated persistent inflammation and a loss of an organized collagen matrix that impairs healing after thermal injury in larval zebrafish (*LeBert et al., 2018*; *LeBert et al., 2015*; *Miskolci et al., 2019*). These features recapitulate human burns and provide an in vivo model system to study the regeneration of sensory neurons in the wound microenvironment.

Using real-time imaging, we took advantage of the optical transparency of larval zebrafish to dissect dynamic cell-cell interactions in the wound microenvironment following injury. We found that localized thermal injury induced axonal damage in the tailfin. Live imaging revealed that sensory axons are physically displaced and experience damage associated with the rapid collective migration of basal keratinocytes following burn. This early keratinocyte migration also contributes to elevated ROS at the tissue scale. Keratinocyte migration was dependent on Arp2/3 signaling and treatment with the Arp2/3 inhibitor CK666 dampened early wound-localized ROS production after burn, suggesting that dysregulated migration perturbs the temporal and spatial organization of ROS after tissue damage. Inhibiting keratinocyte migration via osmotic manipulation with isotonic solution spatially restricted ROS production and rescued sensory axon regrowth and function. Collectively, our results support the importance of regulated keratinocyte behavior for early temporal and spatial signal control that leads to sensory axon regeneration and tissue repair.

## Results

### Burn injury induces peripheral sensory axon damage

To visualize sensory neurons responding to tissue injury, we used 3 days post-fertilization (dpf) *Tg(Ngn1:GFP-Caax)* larval zebrafish that express GFP in sensory neurons (*Andermann et al., 2002*; *Blader et al., 2003*; *McGraw et al., 2008*). Larvae were either mechanically injured by tailfin transection or burn as previously described (*Figure 1A*; *Miskolci et al., 2019*). Intravital imaging of larvae beginning at 24 hr post-wound (hpw) revealed an abnormal axon morphology in burned larvae compared to mechanical transection, with axons showing fewer branch points (*Figure 1B*). To evaluate sensory axons following injury, we assessed axon density in the wounded tissue posterior to the notochord 24 hpw. Larval zebrafish caudal fins can regenerate fully by 3 days post-transection, with 60% of fin regrowth occurring by day 1.5 (*Lisse et al., 2015*). Following transection, axon density was 89.5±0.02% of the density observed in age-matched uninjured larvae 24 hpw. In contrast, we found that burned larvae had significantly reduced sensory axon density, with an axon density of 63.7±0.02% compared to uninjured fins (*Figure 1C*). This relative decrease was sustained even 96 hpw with an axon density of only 65.1±0.04% compared to control (*Figure 1C*). To test whether this regenerative defect was associated with a defect in sensory neuron function, we assessed the touch responsiveness of wounded tissue. Light pressure was applied by an eyelash brush directly to the wound area, and sensory neuron function was scored by the presence of a tail flick reflex (*Granato et al., 1996*). As expected, larvae wounded by transection had a nearly 100% response rate 24 hpw, indicating the rapid recovery of sensory neurons following mechanical injury (*Figure 1D*). In contrast, none of the

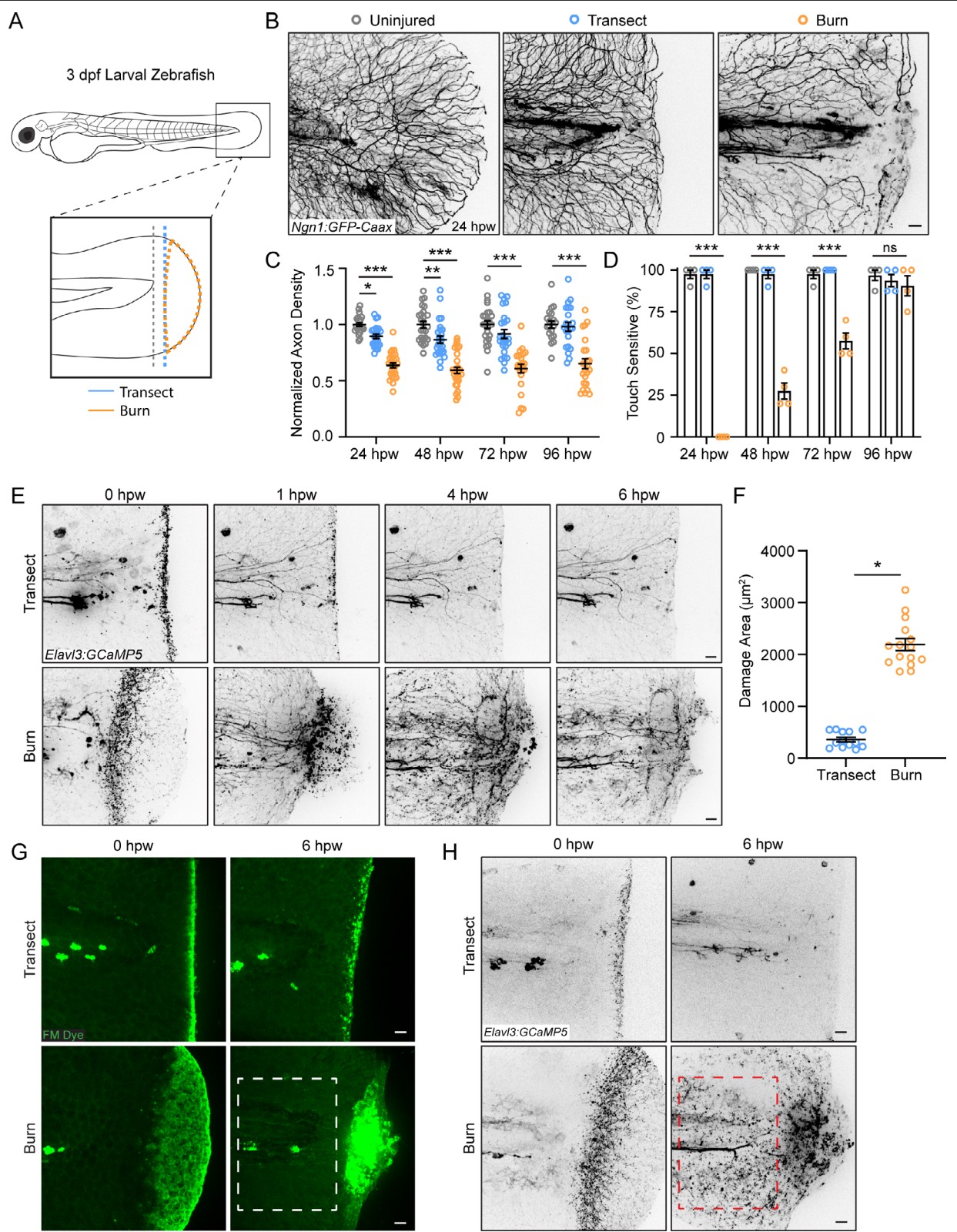

**Figure 1.** Peripheral sensory axons have impaired regeneration after burn injury. (**A**) Schematic of larval zebrafish injury. Gray dashed line denotes area used to measure axon density to the right of the notochord. (**B**) Confocal max-projected images of sensory axons in uninjured, transected, and burned *Tg(Ngn1:GFP-Caax)* caudal fins 24 hr post-wound (hpw). (**C**) Quantification of axon density for uninjured, transected, and burned larvae in the wound area 24–96 hpw. N>20 larvae per condition from four replicates. (**D**) Quantification of sensory perception for uninjured, transected, and burned

*Figure 1 continued on next page*

*Figure 1 continued*

larvae 24–96 hpw. N>32 larvae per condition from four replicates. (**E**) Confocal time-series images of axonal damage, indicated by calcium-positive punctae (black dots), in *Tg(Elavl3:GCaMP5)* larvae following either transection or burn injury. Each series follows one representative larva over 6 hpw. (**F**) Quantification of axon damage area in transected and burned larvae 6 hpw. N>12 larvae per condition from two replicates. (**G**) Images of larvae either transected or burned in the presence of FM 1–43 dye. White dashed box denotes area of uninjured tissue in which axonal damage appears in H. (**H**) Images show axonal damage following transection or burn injury. Red dashed box corresponds to the tissue region highlighted in G. In all cases, scale bars = 20 µm. *p<0.05, **p<0.01, ***p<0.001, ns = not significant.

The online version of this article includes the following source data and figure supplement(s) for figure 1:

**Source data 1.** Numerical data for *Figure 1C*.

**Source data 2.** Numerical data for *Figure 1D*.

**Source data 3.** Numerical data for *Figure 1F*.

**Figure supplement 1.** Elavl3-GCaMP5 transgenic fish show sensory axon damage.

burned larvae were sensitive to touch 24 hpw with resolution occurring by 96 hpw (*Figure 1D*). Importantly, all tested larvae exhibited a tail flick reflex when pressure was applied to the trunk, showing that the impaired sensation was limited to the damaged tissue.

We next sought to further investigate sensory neuron function in burned tissue. For this, we assessed wound-induced axonal damage using *Tg(Elavl3:GCaMP5)* zebrafish larvae that express the calcium probe GCaMP under the pan-neuronal Elavl3 promoter. Using these larvae, we observed no noticeable change in GCaMP intensity under homeostatic conditions (*Figure 1—figure supplement 1A*). While transient calcium increase following cell damage is required for immediate membrane repair and subsequent regeneration, chronically elevated cytosolic calcium is associated with cell degeneration and death (*Khaitin, 2021*). GCaMP has previously been used as a marker of real-time axonal damage in zebrafish (*Adalbert et al., 2012*; *Linsley et al., 2021*; *Ziv and Spira, 1993*). Therefore, GCaMP can be used for real-time labeling of axon damage in larval zebrafish (*Adalbert et al., 2012*; *Linsley et al., 2021*; *Ziv and Spira, 1993*). Axon damage is characterized by the fragmenting of axons and the formation of small punctae, which are later cleared by phagocytes (*Reyes et al., 2004*; *Villegas et al., 2012*). Accordingly, the neurotoxin sodium azide elicited widespread and long-lasting calcium-positive punctae, indicating sustained axonal damage (*Figure 1—figure supplement 1A*; *Linsley et al., 2021*). Unlike Ngn1, the Elavl3 promoter is expressed by both sensory and motor neurons. To ensure that the calcium increase in wounded tissue was specific to sensory axons, sensory neurons were depleted by injecting a morpholino targeting Ngn1 (*Cornell and Eisen, 2002*). These larvae did not have any obvious developmental defects but lacked responsiveness to touch stimulation, as previously reported (*Cornell and Eisen, 2002*). As expected, no calcium increase was detected in Ngn1 depleted larvae following injury (*Figure 1—figure supplement 1B*).

Time-lapse images were taken of Elavl3:GCaMP5 larvae to visually capture instances of axon damage indicated by calcium-positive punctae during the 30 min after injury (*Figure 1—figure supplement 1C*). These images were compared to time-lapse images of sensory neuron-labeled larvae (*Figure 1—figure supplement 1D*). In agreement with previous observations of axonal damage following mechanical injury (*Arrázola et al., 2019*; *Rasmussen et al., 2015*; *Rieger and Sagasti, 2011*), tailfin transection resulted in spatially localized axonal damage that was almost completely resolved by 1 hpw (*Figure 1E*). In contrast, burn injury resulted in a distinct temporal and spatial profile of sensory axon damage. While initial wound-induced sensory neuron-specific calcium increase appeared to be localized to burned tissue, axonal damage continued to increase and spread across the tissue for approximately 6 hr (*Figure 1E and F*). This raised the question of whether axonal damage was restricted to epithelial tissue directly impacted by injury, as observed in transected larvae. To label wounded epithelium, we used the lipophilic dye FM 1–43, which is commonly used to label damaged cell membranes after wounding (*Defour et al., 2014*; *Sønder et al., 2022*; *McDade et al., 2021*; *Shannon et al., 2017*). Immediately following either transection or thermal injury, axonal damage overlapped spatially with wounded epithelial tissue (*Figure 1G and H*). However, by 6 hr following burn injury, there was widespread damage to axons that extended beyond the initial wound area (*Figure 1G and H*). These findings suggest that burn injury induces axonal damage that accumulates over time and is spatially uncoupled from the surrounding epithelial damage.

## The burn wound microenvironment contributes to defective axon regeneration and function

To determine if early wound signaling regulates sensory axon regeneration, we used a two-wound model (*Figure 2A*) to excise the burned tissue at different times post-wound. In this system, zebrafish were first injured by either a primary tailfin transection or burn, and then a secondary transection injury was carried out either early at 5 min post-wound (mpw) or late at 6 hpw. The secondary transection was performed so that all the burned tissue was excised (*Figure 2A and B*). As expected, larvae that underwent only transections at both time points had full sensory function by 24 hpw regardless of the timing of the second transection injury, showing that zebrafish efficiently heal after mechanical damage (*Figure 2C–E*). In burned larvae, early transection after thermal injury (5 mpw) improved sensory axon regeneration and function to levels similar to transected larvae (*Figure 2C–E*), suggesting that burn injury does not immediately affect sensory axons differently than mechanical damage. However, when burned tissue was excised after 6 hr, significant defects were noted in both axon density and sensory function compared to larvae that either received two transection injuries or had burned tissue transected at 5 mpw (*Figure 2C–E*). These findings suggest that the local wound environment modulates sensory axon outcomes following burn injury, and that events in the first 6 hr after injury impact longer-term sensory axon repair.

## Burn injury induces the early collective movement of keratinocytes and sensory axons

To understand how burn injury damages sensory axons, we performed live imaging of epithelial keratinocytes, which closely associate with sensory axons (*O'Brien et al., 2012*; *Rieger and Sagasti, 2011*; *Rosa et al., 2023*). Live imaging of *Tg(Krt4:UtrCH-GFP)* larvae that express the actin probe Utrophin under a pan-keratinocyte promoter allowed for visualization of keratinocyte dynamics following either mechanical or burn injury. In response to transection, we initially observed characteristic epithelial cell contraction at the wound edge; however, keratinocytes distal to the wound edge remained relatively stationary (*Figure 3A*; *Video 1*). In contrast, burn wounding resulted in a rapid collective movement of keratinocytes toward the site of tissue damage (*Figure 3A*). To quantify the rapid movement of keratinocytes in burn injured larvae, we used *Tg(Krtt1c19e:acGFP)* zebrafish to specifically label motile basal epithelial cells (*Lee et al., 2014*). Live-imaging experiments revealed that basal keratinocytes, on average, moved a total distance of 205.7±10.7 µm in the first hour following burn injury, which was significantly greater than the 58.9±5.8 µm of migration observed following tailfin transection (*Figure 3B*; *Figure 3—figure supplement 1A*). Although keratinocytes moved as a collective in response to burn injury, they exhibited chaotic movement and appeared to be loosely associated with their neighbors.

To determine if this process was due to active migration, time-lapse imaging was performed using *Tg(Krt4:Lifact-mRuby)* larvae to visualize actin dynamics of both the superficial and basal layers. Unwounded larvae had regularly shaped keratinocytes with even actin distribution around the cell periphery in both the superficial and basal layers (*Figure 3C*), indicative of a non-migratory epithelium. Superficial keratinocytes in burned larvae were elongated and had an even actin distribution, but the basal keratinocytes showed actin localization to the leading edge with the formation of lamellipodia (*Figure 3C*), suggesting that the superficial keratinocytes are being pulled by the motile basal keratinocytes.

The axons of sensory neurons are ensheathed within actin-rich channels running through basal keratinocytes (*O'Brien et al., 2012*; *Jiang et al., 2019*). Given the chaotic and sustained keratinocyte migration associated with burn injury, we next tested if sensory axons are displaced along with associated migrating keratinocytes. Simultaneous imaging of basal keratinocytes and sensory axons following thermal injury revealed that sensory axon movement is coordinated with keratinocyte migration (*Figure 3D*; *Video 2*). To characterize the kinetics of axonal damage following burn, time-lapse movies were performed with *Tg(Elavl3:GCaMP5)* larvae to determine if the onset of axonal damage occurs during basal keratinocyte migration. Within the first hour after burn, calcium-positive punctae were identified that coincided with keratinocyte migration, indicating that early keratinocyte migration was associated with the initial axonal damage (*Video 3*). To determine if damage was limited to axons, we imaged the cell bodies of peripheral sensory neurons. At 3 dpf, the skin is innervated by Rohon-Beard (RB) and dorsal root ganglia (DRG) neurons with cell bodies that reside within and

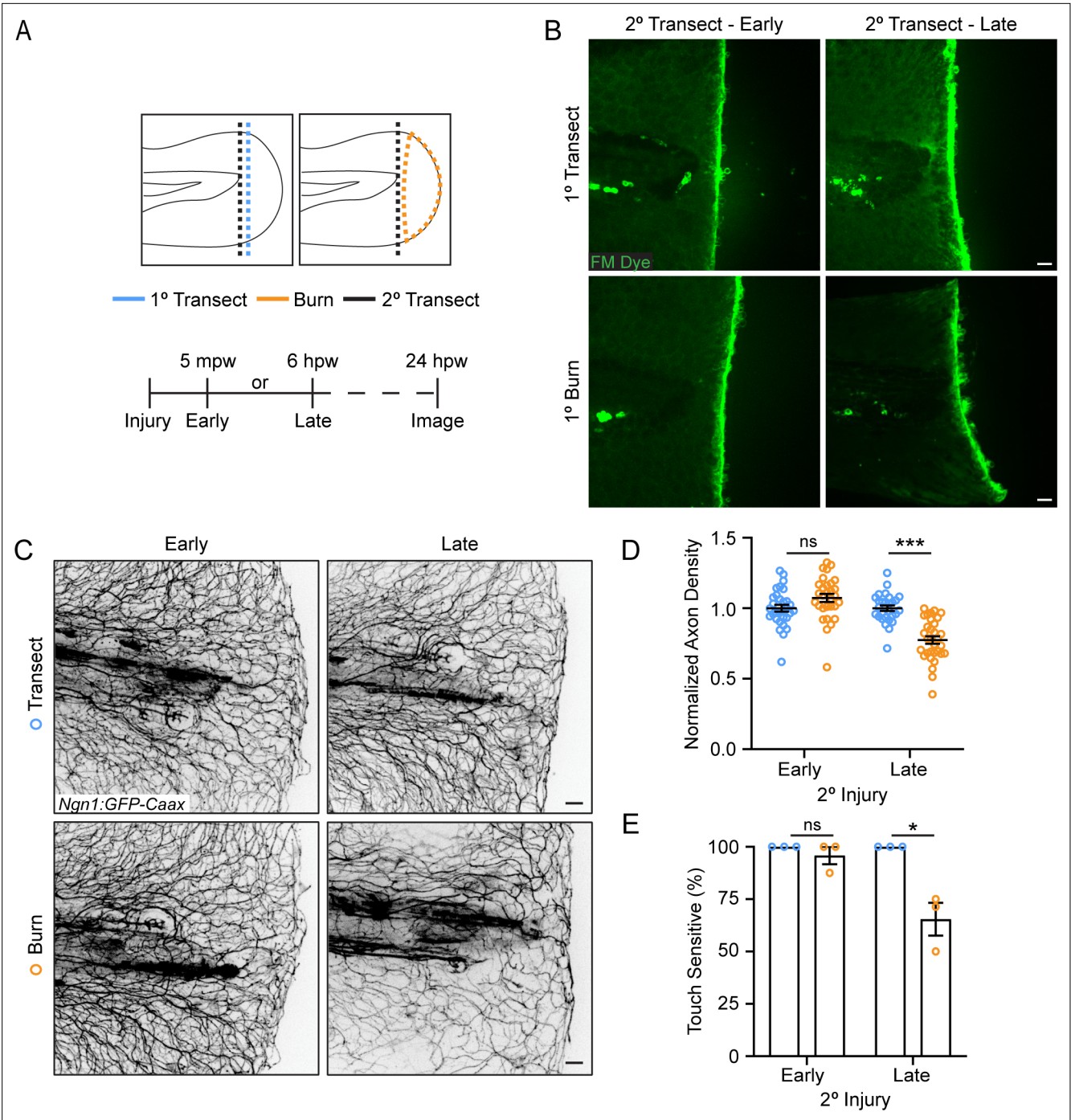

**Figure 2.** The burn wound microenvironment contributes to impaired sensory axon regeneration. (**A**) Schematic of two-wound experiment design. (**B**) Confocal max-projected images of FM dye staining following secondary transection in the two-wound experiment at 5 min post-wound (mpw) and 6 hr post-wound (hpw). (**C**) Images of sensory axons in larvae subjected to an initial transection or burn injury followed by subsequent transection either early (5 mpw) or late (6 hpw). (**D**) Quantification of axon density in wounded tissue 24 hpw from larvae wounded as in B. N>28 larvae per condition from three replicates. (**E**) Quantification of sensory perception in wounded tissue 24 hpw from larvae wounded as in B. N=24 larvae each from three replicates. In all cases, scale bars = 20 μm. *p<0.05, ***p<0.001, ns = not significant.

The online version of this article includes the following source data for figure 2:

**Source data 1.** Numerical data for *Figure 2D*.

**Source data 2.** Numerical data for *Figure 2E*.

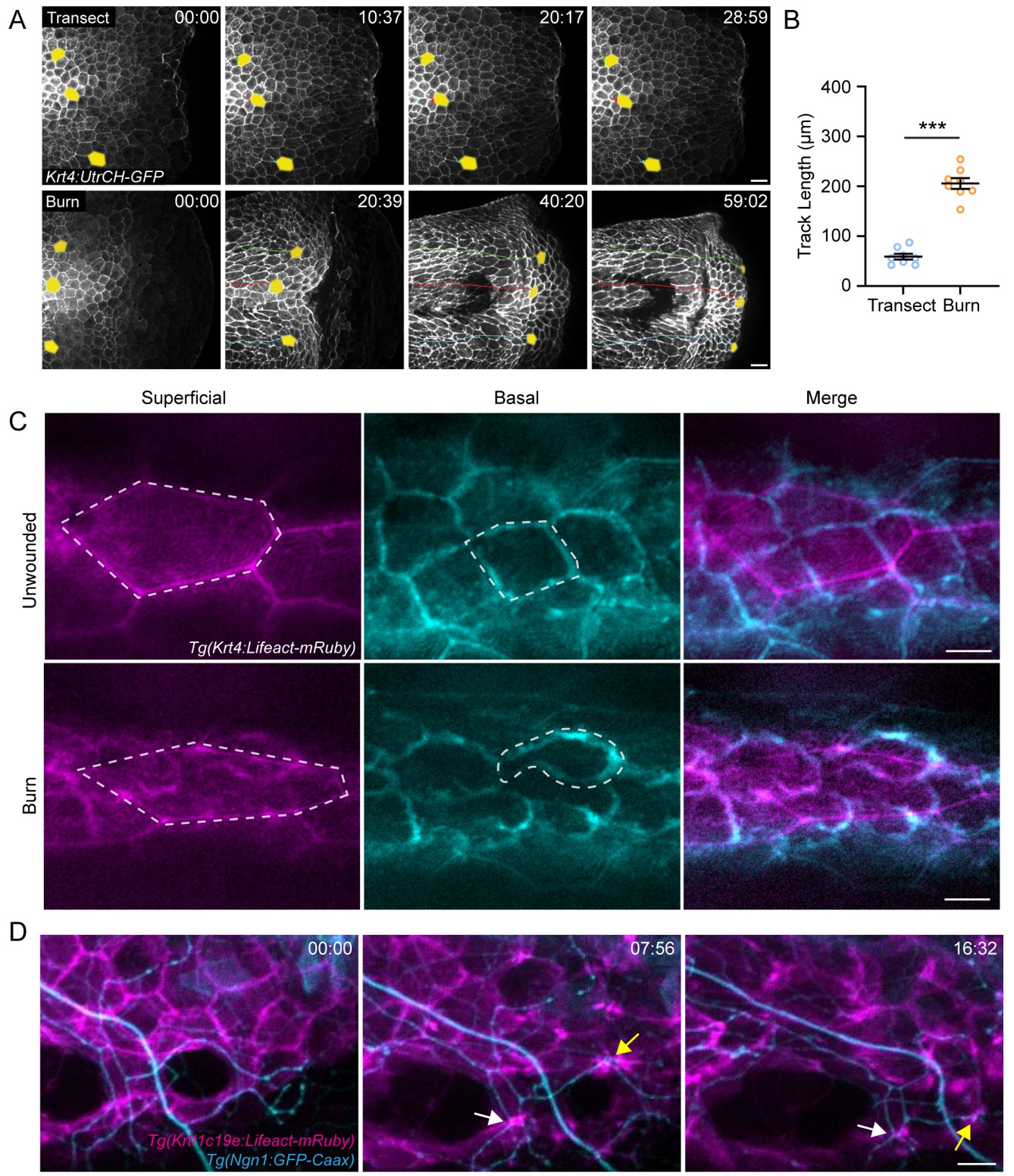

**Figure 3.** Burn injury induces coordinated keratinocyte and sensory axon movement. (**A**) Confocal max-projected time-series images of *Tg(Krt4:UtrCH-GFP)* larvae after either transection or burn injury. Yellow pseudocolored cells and colored tracks highlight keratinocyte displacement. Scale bar = 20 μm. (**B**) Quantification of keratinocyte movement distance over 1 hr post-wound (hpw). N = 8 larvae each collected from three replicates. (**C**) Confocal max-projected images of superficial and basal keratinocytes in *Tg(Krt4:Lifeact-mRuby)* labeled larvae. Left, superficial keratinocytes. Middle, basal keratinocytes. Right, merge. Superficial and basal cell images were taken from the same z-stack and pseudocolored to match the appropriate cell

*Figure 3 continued on next page*

*Figure 3 continued*

layer. Dashed lines outline one individual keratinocyte. Scale bar = 10 µm. (**D**) Confocal max-projected time-series images of sensory axons and basal keratinocytes in dual-labeled *Tg(Krt4:Lifeact-mRuby); Tg(Ngn1:GFP-Caax)* larvae unwounded or after burn. Arrows highlight coincident movement between keratinocytes and associated sensory axons. Unless otherwise stated, scale bar = 20 µm. \*\*\*p<0.001.

The online version of this article includes the following source data and figure supplement(s) for figure 3:

**Source data 1.** Numerical data for *Figure 3A*.

**Figure supplement 1.** Basal keratinocyte migration in response to injury.

**Figure supplement 2.** Sensory axon cell bodies are not displaced following burn injury.

just outside the spinal cord, respectively. Following burn injury, we noted that both the RB and DRG cell bodies that innervate the caudal fin were intact and non-motile in comparison to their pre-burn morphology (*Figure 3—figure supplement 2A and B*). Taken together, these findings indicate that sensory axons are displaced along with the collective movement of basal keratinocytes following burn, and that this early movement is associated with the start of axonal damage.

## The Arp 2/3 inhibitor CK666 impairs initial basal keratinocyte migration and modulates early ROS signaling following burn

We next determined if a known regulator of leading-edge actin dynamics and migration, Arp2/3, modulates the movement of basal keratinocytes after burn (*Henson et al., 2015*). Treatment with the Arp2/3 inhibitor CK666 limited keratinocyte lamellipodia formation and impaired early keratinocyte migration, indicating that the early keratinocyte movement is Arp2/3 dependent (*Figure 4A–C*). Although early migration was impaired, by 40 mpw the migration was not significantly different between control and CK666-treated larvae, suggesting that CK666 treatment only inhibits early keratinocyte migration. To determine if this early treatment altered signaling in wounded tissue, we probed the effects of CK666 treatment on the generation of ROS signaling at the burn wounds. It is known that efficient tissue repair after injury relies on coordinated ROS production by epithelial cells (*Enyedi and Niethammer, 2015*; *Jelcic et al., 2019*; *Yoo et al., 2012*). Following mechanical injuries such as tailfin transection or laser ablation, transient and localized $H_2O_2$ also promotes sensory axon regeneration and wound healing (*Cadiz Diaz et al., 2022*; *Rieger and Sagasti, 2011*). Because of this ROS requirement, we hypothesized that the robust keratinocyte movement and sustained damage in burned tissue may result in dysregulated ROS production. To test this, $H_2O_2$ level was determined using the fluorescent dye pentafluorobenzenesulfonyl fluorescein (Pfbsf), an established readout of ROS production that has previously been used in larval zebrafish (*Maeda et al., 2004*; *Niethammer et al., 2009*). Early after burn wounding, there

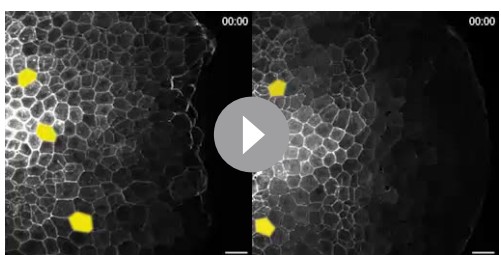

**Video 1.** Burn injury induces keratinocyte movement. *Tg(Krt4:UtrCH-GFP)* larvae were injured either by tailfin transection (left) or burn (right). While minimal keratinocyte movement is observed following transection, burn injury results in keratinocyte movement toward the wound edge for approximately 1 hr post-wound. Yellow pseudocolored cells indicate representative keratinocyte movement. Images were collected at 2 frames/min. Scale bar = 20 µm.
https://elifesciences.org/articles/94995/figures#video1

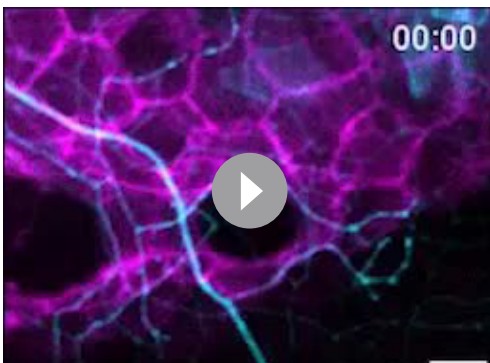

**Video 2.** Sensory axons move with associated keratinocytes following injury. Basal keratinocyte (magenta), *Tg(Krtt1c19e:Lifeact-Ruby)*, and sensory axon (cyan), *Tg(Ngn1:GFP-Caax)*, movement was tracked following burn injury. Arrows highlight regions where keratinocyte and sensory axon movement is spatially coincident. Images were collected at 3 frames/min. Scale bar = 10 µm.
https://elifesciences.org/articles/94995/figures#video2

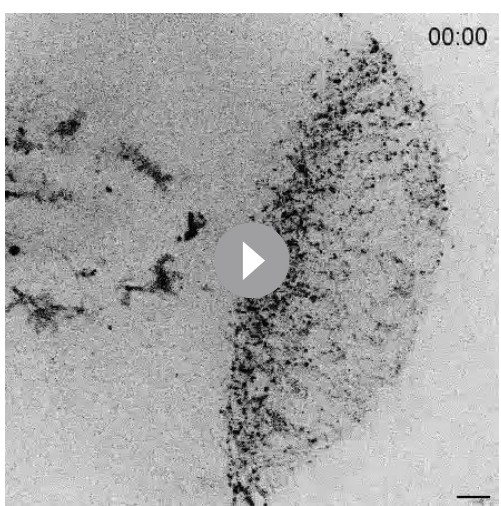

**Video 3.** Tissue movement is associated with axonal damage following burn injury. *Tg(Elavl3:GCaMP5)* larva was burn wounded to track axonal damage, indicated by elevated intracellular calcium (black dots). Damage present at time 0 min is due to the burn wound itself, while new axonal damage coincides with keratinocyte movement. Images collected at 2 frames/min. Scale bar = 20 µm.

https://elifesciences.org/articles/94995/figures#video3

was robust generation of hydrogen peroxide in burned tissue that was dampened in CK666-treated larvae. In the presence of CK666, $H_2O_2$ production was concentrated at the wound edge, similar to what has been reported with tail transection (*Figure 4D and E*). These findings suggest that early migration alters the temporal and spatial distribution of ROS after wounding. Interestingly, CK666-treated larvae had no significant difference in axon damage or regeneration 24 hr after burn compared to controls, although there was a trend toward improved sensory function (*Figure 4F–H*). Taken together, the findings suggest that keratinocyte migration regulates early tissue scale ROS production after burn injury.

The direct effects of early ROS signaling on burn wound healing were also tested using the drug diphenyleneiodonium (DPI) to inhibit ROS production (*Li and Trush, 1998*; *Niethammer et al., 2009*; *Yoo et al., 2012*). Treatment longer than 1 hpw was not possible due to toxicity. With this short treatment, larvae showed some improvement in axon density and touch sensitivity, although it was not statistically significant (*Figure 4—figure supplement 1A–C*). These data suggest that early ROS inhibition is not sufficient to rescue axon regeneration and function, although it is possible that short-term dampening of ROS may improve sensory neuron function.

## Isotonic solution limits keratinocyte movement induced by burn injury and alters the temporal and spatial distribution of redox signaling

Previous studies have demonstrated that the presence of an osmotic gradient promotes keratinocyte migration via cell swelling in response to mechanical injury (*Gault et al., 2014*). To determine if other treatments that affect keratinocyte migration also impact tissue scale ROS production after burn, we determined if altering osmotic balance impacts keratinocyte migration and the distribution of ROS signaling after burn. Under control conditions, zebrafish are maintained in hypotonic solution. Removing this osmotic gradient by wounding larvae in the presence of solution that is isotonic to the interstitial fluid has previously been shown to inhibit keratinocyte migration following mechanical injury and impair wound healing (*Gault et al., 2014*; *Kennard and Theriot, 2020*). We found that wounding in the presence of an isotonic solution prevented the rapid movement of keratinocytes in response to a burn. Within the first hour following burn injury, keratinocyte average speed was reduced from 0.059 µm/s in control medium to 0.003 µm/s in isotonic medium (*Figure 5A–C*; *Video 4*). Immediately following burn injury, the level of $H_2O_2$ was the same between the treatment groups, indicating that cells wounded in the presence of isotonic solution maintain their normal ability to generate ROS (*Figure 5F and G*; *Enyedi and Niethammer, 2015*). Examining ROS production during the first hour post-burn revealed that control larvae had increased ROS throughout the fin tissue compared to isotonic-treated larvae, while both conditions had comparable levels of ROS at the wound (*Figure 5D and E*). At 6 hr post-burn, $H_2O_2$ production was no longer localized to the wound edge in control burned larvae and had increased throughout the tailfin. By contrast, $H_2O_2$ remained restricted to the wound edge in larvae burned in the presence of isotonic solution, displaying a similar localized pattern to that observed after mechanical injury (*Figure 5F*; *Jelcic et al., 2019*; *Korte et al., 2022*). Quantification revealed that $H_2O_2$ level at the wound edge was similar between control and isotonic-treated larvae 6 hpw. However, the level of $H_2O_2$ was approximately sixfold lower in the fin epithelial tissue adjacent to the burn wound with isotonic treatment (*Figure 5F and H*). These findings

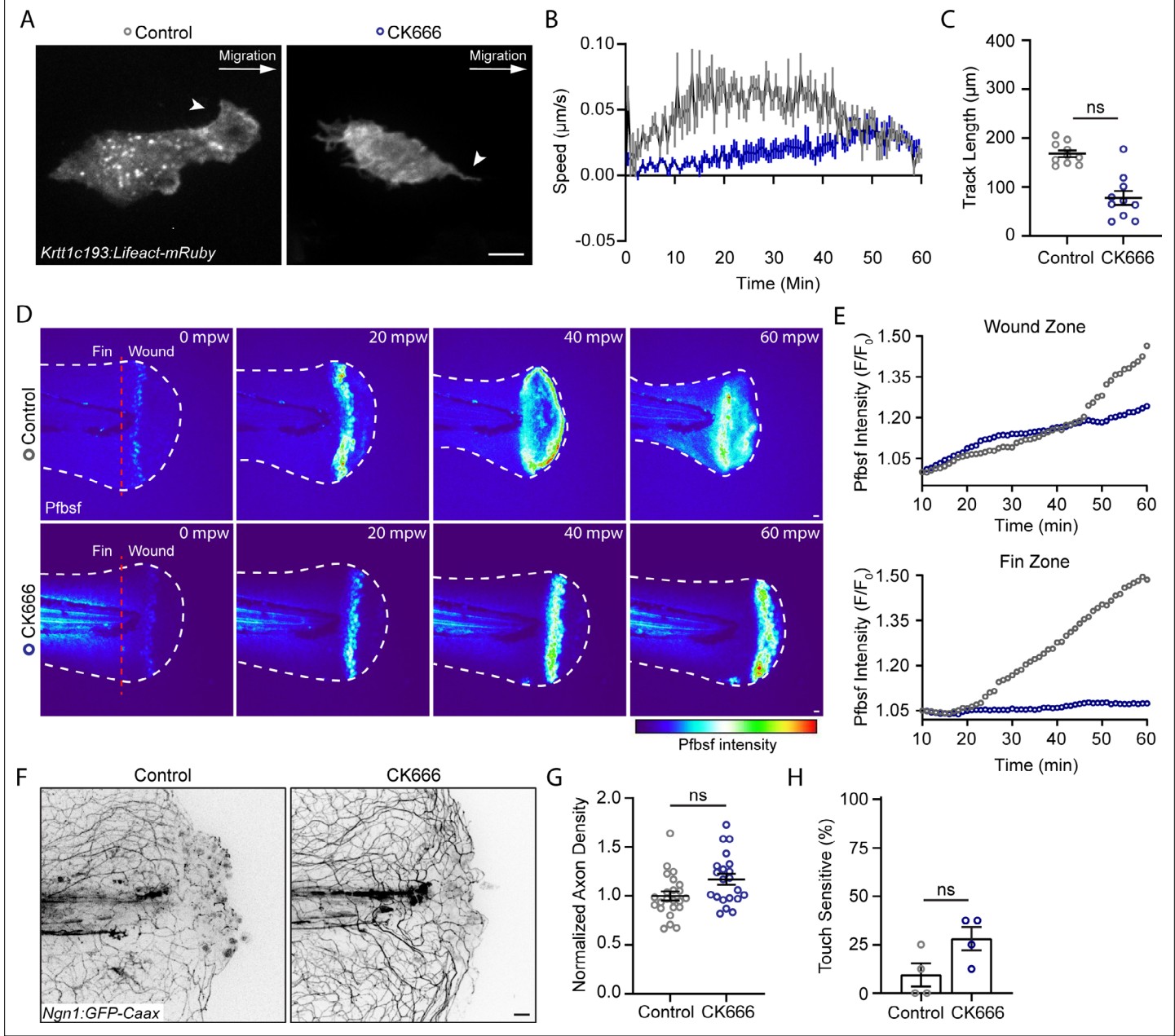

**Figure 4.** The Arp 2/3 inhibitor CK666 impairs early keratinocyte movement and alters the spatial distribution of reactive oxygen species signaling. (**A**) Confocal max-projected images of control or CK666-treated transiently injected *Tg(Krtt1c19e:Lifeact-mRuby)* larvae. Arrows point to lamellipodia in the control larva, and lack of lamellipodia in the CK666-treated larva. Scale bar = 10 µm. (**B**) Plot of keratinocyte speed over 1 hr post-wound (hpw) as treated in A. N=10 larvae each collected from three replicates. (**C**) Plot of keratinocyte distance moved over 1 hpw as treated in A. N=10 larvae each collected from three replicates. (**D**) Confocal sum-projected time-series images of hydrogen peroxide level (pentafluorobenzenesulfonyl fluorescein [Pfbsf] intensity) in 1 larva over 1 hpw in the indicated treatment. (**E**) Quantification of Pfbsf intensity in the wound or fin area of the represented larva after burn injury as treated in D over 1 hpw. N=1 representative larva per condition. (**F**) Confocal max-projected images of sensory axons 24 hpw in larvae wounded in control medium or CK666. (**G**) Quantification of axon density 24 hpw in larvae treated as in J. N>22 larvae per condition from four replicates. (**H**) Quantification of sensory perception 24 hpw in larvae treated as in J. N=32 larvae per condition from four replicates. Unless otherwise specified, scale bars = 20 µm. ns = not significant.

The online version of this article includes the following source data and figure supplement(s) for figure 4:

**Source data 1.** Numerical data for *Figure 4B*.

**Source data 2.** Numerical data for *Figure 4C*.

**Source data 3.** Numerical data for *Figure 4E*.

*Figure 4 continued on next page*

*Figure 4 continued*

**Source data 4.** Numerical data for *Figure 4G*.

**Source data 5.** Numerical data for *Figure 4H*.

**Figure supplement 1.** Early reactive oxygen species (ROS) inhibition is not sufficient to improve axon regeneration.

**Figure supplement 1—source data 1.** Numerical data for *Figure 4—figure supplement 1B*.

**Figure supplement 1—source data 2.** Numerical data for *Figure 4—figure supplement 1C*.

suggest that robust keratinocyte movement induced by burn injury generates an oxidative environment at the tissue scale.

## Isotonic medium is sufficient to improve sensory neuron regeneration and function after burn

We next assessed sensory axon damage after burning in isotonic solution. Immediately following injury, larvae burned in isotonic solution displayed axon damage similar to larvae injured in control medium (*Figure 6A*). However, 6 hpw, axonal damage in isotonic-treated larvae was reduced and remained restricted to the site of injury, similar to the spatially restricted $H_2O_2$ signal induced by isotonic treatment (*Figure 6A and B*). Accordingly, larvae burned in isotonic medium had significantly greater axon density 24 hpw, and more than 85% of isotonic-treated larvae had normal sensory function by 24 hpw (*Figure 6C–E*). To determine if the benefit of isotonic solution was due to its ionic composition, we tested the effects of an isotonic solution of the sugar D-Sorbitol. We also found that an isotonic solution with D-Sorbitol limited basal keratinocyte migration and had normal axon density and sensory function 24 hpw, suggesting that the benefit of isotonic solution is independent of effects on electrical cues (*Figure 6—figure supplement 1A–C*).

To determine if early keratinocyte migration contributes to the impact of isotonic solution on sensory neuron function at later time points, we treated with isotonic solution starting 1 hr following burn injury (*Figure 6F*). When isotonic medium was added 1 hpw, after keratinocyte migration was complete, there was no rescue of ROS production in either the wound area or the fin 6 hpw (*Figure 6G and H*). Additionally, there was no improvement in sensory axon density or function 24 hpw, supporting the idea that early wound events during the first hour are critical for their effects on later sensory neuron function (*Figure 6I–K*). Collectively, these findings suggest that early keratinocyte movement after burn coordinates spatial redox signaling and impacts sensory axon regeneration.

## Discussion

Tissue repair requires the coordination of signaling across spatial and temporal scales. Our prior work has shown that early ROS signaling immediately after mechanical damage is necessary for longer-term tissue repair (*Yoo et al., 2012*). Wound-induced ROS production is also required for leukocyte recruitment, ECM remodeling, and sensory axon regeneration in response to tissue injury (*LeBert et al., 2018*; *Niethammer et al., 2009*; *Rieger and Sagasti, 2011*; *Yoo et al., 2011*). While the requirement of ROS production following tissue injury is clear, we lack an understanding of how early redox signaling is coordinated temporally and spatially to mediate long-term tissue repair. We recently reported that burn injury induces a distinct repair response with impaired collagen remodeling and delayed healing (*LeBert et al., 2018*; *Miskolci et al., 2019*). In light of the known defect in sensory function after burn injuries in humans, we sought to determine how the early epithelial response modulates sensory axon recovery. Our findings suggest that early damage-induced keratinocyte movement plays a role in the spatial patterning of ROS production in the wound microenvironment and impacts the ability of sensory axons to regenerate.

Collective keratinocyte migration is conserved across species and is required to mediate wound closure after tissue injury (*Kirfel and Herzog, 2004*; *Mayor and Etienne-Manneville, 2016*). While collective cell migration has been observed in larval zebrafish previously (*Olson and Nechiporuk, 2021*; *Yamaguchi et al., 2022*), its contribution to tissue repair remains unclear. In comparison to the organized movement associated with keratinocyte response to mechanical injury, our observations here identify excessive keratinocyte migration as a defining feature of the response to burn injury. Basal keratinocyte migration following burn injury appeared to lack a stereotypical leader-follower

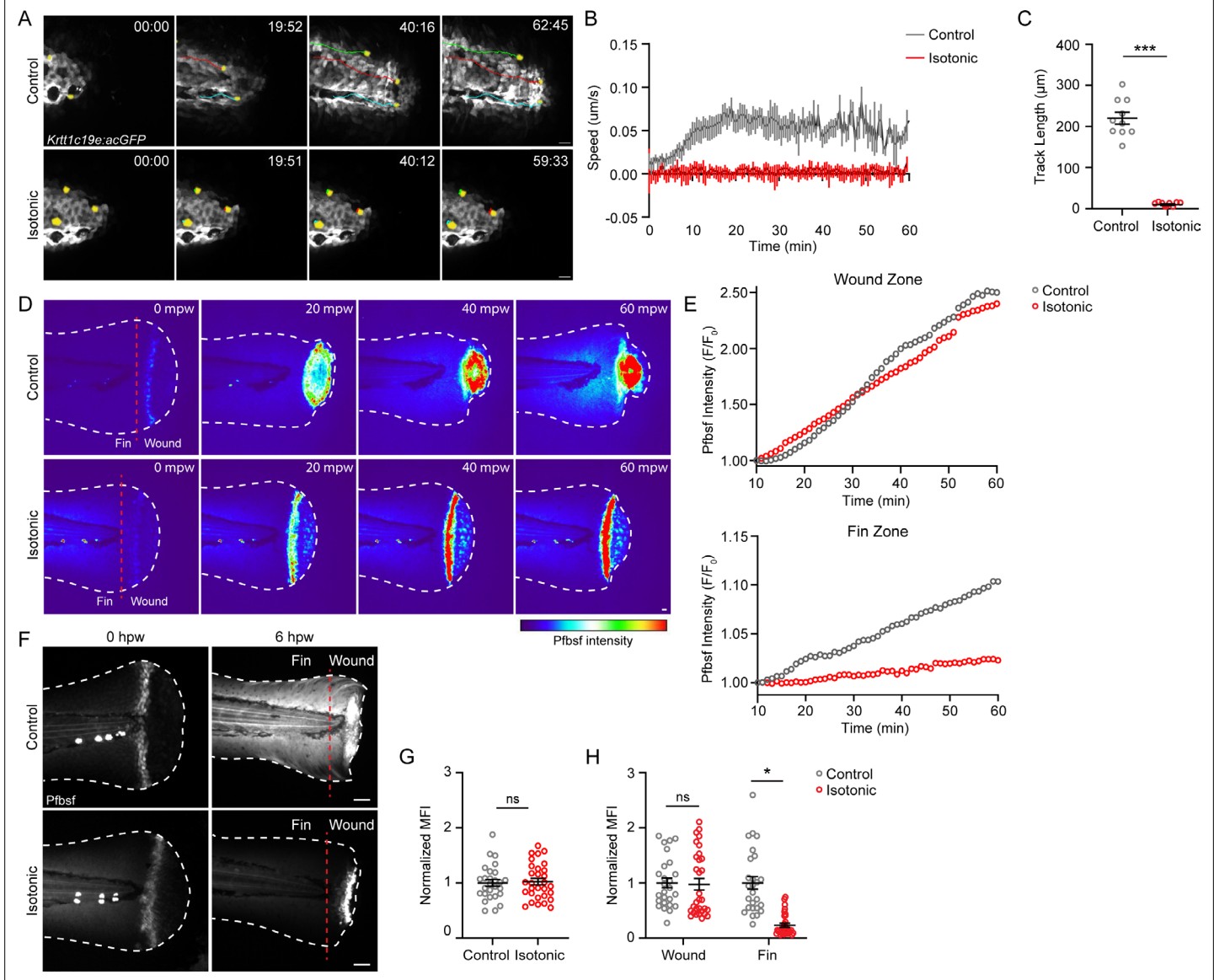

**Figure 5.** Treatment with isotonic solution inhibits keratinocyte migration and dampens reactive oxygen species (ROS) signaling. (**A**) Confocal time-series images of basal keratinocyte movement in *Tg(Krtt1c19e:acGFP)* larvae over 1 hr post-wound (hpw) after burn injury in the indicated treatment. (**B**) Plot of basal keratinocyte average speed over 1 hpw treated as in A. N=10 larvae per condition collected from three replicates. (**C**) Distance of keratinocyte movement over 1 hpw treated as in A. N=10 larvae per condition collected from three replicates. (**D**) Confocal sum-projected, heat-mapped time-series images of hydrogen peroxide level (pentafluorobenzenesulfonyl fluorescein [Pfbsf] intensity) over 1 hpw as treated in A. (**E**) Quantification of Pfbsf intensity in the wound or fin area of the represented larva after burn injury as treated in D over 1 hpw. N=1 representative larva per condition. (**F**) Confocal sum-projected images of Pfbsf intensity in the fin and wound zone either 0 or 6 hr following burn injury. Dashed red line denotes the boundary between the wound area and distal fin tissue. Scale bar = 50 µm. (**G**) Quantification of mean Pfbsf fluorescence intensity (MFI) immediately (0 hpw) after burn injury normalized to the control condition. N>27 larvae per condition from three replicates. (**H**) Quantification of MFI 6 hpw in the indicated region of the fin normalized to the control condition. N>26 larvae per condition from three replicates. Unless otherwise indicated, scale bars = 20 µM. *p<0.05, ***p<0.001, ns = not significant.

The online version of this article includes the following source data for figure 5:

**Source data 1.** Numerical data for *Figure 5B*.

**Source data 2.** Numerical data for *Figure 5C*.

**Source data 3.** Numerical data for *Figure 5E*.

**Source data 4.** Numerical data for *Figure 5G*.

**Source data 5.** Numerical data for *Figure 5H*.

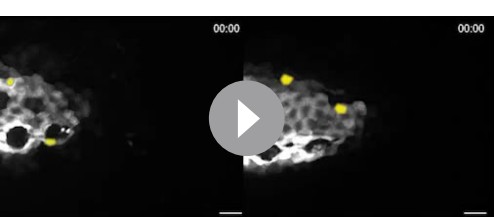

**Video 4.** Wounding in isotonic medium prevents burn-induced keratinocyte movement. Basal keratinocyte, *Tg(Krtt1c19e:acGFP)*, movement was tracked in control (left) and isotonic-treated (right) larvae following burn injury. Wounding in the presence of isotonic medium prevents keratinocyte movement associated with burn wounding. Yellow pseudocolored cells indicate representative keratinocyte movement. Images were collected at 2 frames/min. Scale=20 µm.

https://elifesciences.org/articles/94995/figures#video4

dynamic, with cells instead moving independently of one another but as a collective group. This observation suggests that collective keratinocyte migration is a feature of tissue repair in larval zebrafish regardless of the mode of injury and that its regulation is required for the success of long-term healing. Indeed, we provide evidence that early migration and formation of lamellipodia in basal keratinocytes requires Arp2/3 signaling, and that this aberrant migration regulates the temporal and spatial distribution of early redox signaling in the wound tissue. However, Arp2/3 inhibition did not lead to sustained control of keratinocyte migration and production of ROS eventually increased throughout the tailfin. Accordingly, sensory function was not significantly improved at 24 hpw.

To further modulate keratinocyte movement, we took advantage of isotonic treatment that is known to affect keratinocyte migration after wounding by tailfin transection. Cell swelling is thought to induce migration by promoting branched actin polymerization and lamellipodia formation (*Bera et al., 2022*; *Han et al., 2012*; *Sforna et al., 2022*), potentially through the activity of mechanically activated ion channels. Previous groups have shown that osmotic differences trigger both ATP release and lamellipodia formation in basal keratinocytes, which promote keratinocyte migration (*Chen et al., 2019*; *Gault et al., 2014*). Our results also show that osmotic modulation affects basal keratinocyte motility in response to burn injury. Interestingly, limiting keratinocyte motility by isotonic treatment is detrimental to tissue repair following mechanical injury (*Gault et al., 2014*), but isotonic treatment both rescues epithelial morphology and reduces axon damage following thermal injury. This suggests that there is an optimal amount of keratinocyte movement needed for efficient repair and long-term regeneration.

A conceptual challenge in wound repair has been understanding how early wound-induced events are linked to long-term repair (*Sonnemann and Bement, 2011*). ROS signaling provides a framework to understand this link due to its requirement for both early wound contraction and long-term regeneration. In zebrafish, the ROS $H_2O_2$ is generated along a tissue scale gradient with the highest levels at the wound edge (*Niethammer et al., 2009*). While this spatial gradient undoubtedly directs cell function based on the position along the gradient, uncontrolled ROS production is damaging to tissues. Therefore, a mechanism must exist to control ROS such that it remains relatively localized to the site of damage and is controlled temporally and spatially. In addition to their early role in wound resealing, keratinocyte redox signaling is critical for long-term repair. Cell swelling induces cPLA2-dependent 5-oxoETE production and immune cell recruitment (*Enyedi et al., 2013*). This suggests the signals that control keratinocyte motility may simultaneously modify long-term keratinocyte signaling.

Our findings suggest that cell migration can modulate tissue scale signaling following injury. Importantly, isotonic treatment blocks keratinocyte movement and restores localized ROS signaling at the wound edge. The dependence on migration was supported both by the effects of Arp2/3 inhibition on early ROS signaling and by the finding that isotonic treatment started after migration was complete (1 hr after injury) did not rescue tissue scale ROS. This suggests that early keratinocyte migration patterns the tissue scale ROS distribution. Additionally, ROS inhibition with the drug DPI provided only slight benefits to axon regeneration. This could be due to treatment being sustained for only 1 hpw, but migration was also not inhibited which further points to the importance of migration in early wound signaling. The finding that isotonic treatment at the time of injury was sufficient to rescue sensory function but treatment after 1 hr did not rescue axon regeneration further highlights the importance of this early motile response for setting up the longer-term repair after damage.

The benefit of a system in which keratinocyte motility controls downstream signaling is twofold. First, it enables signaling to be scaled to the size of injury. If more cells migrate due to a larger injury, then production of ROS will likewise increase. Second, this system provides a mechanism to control

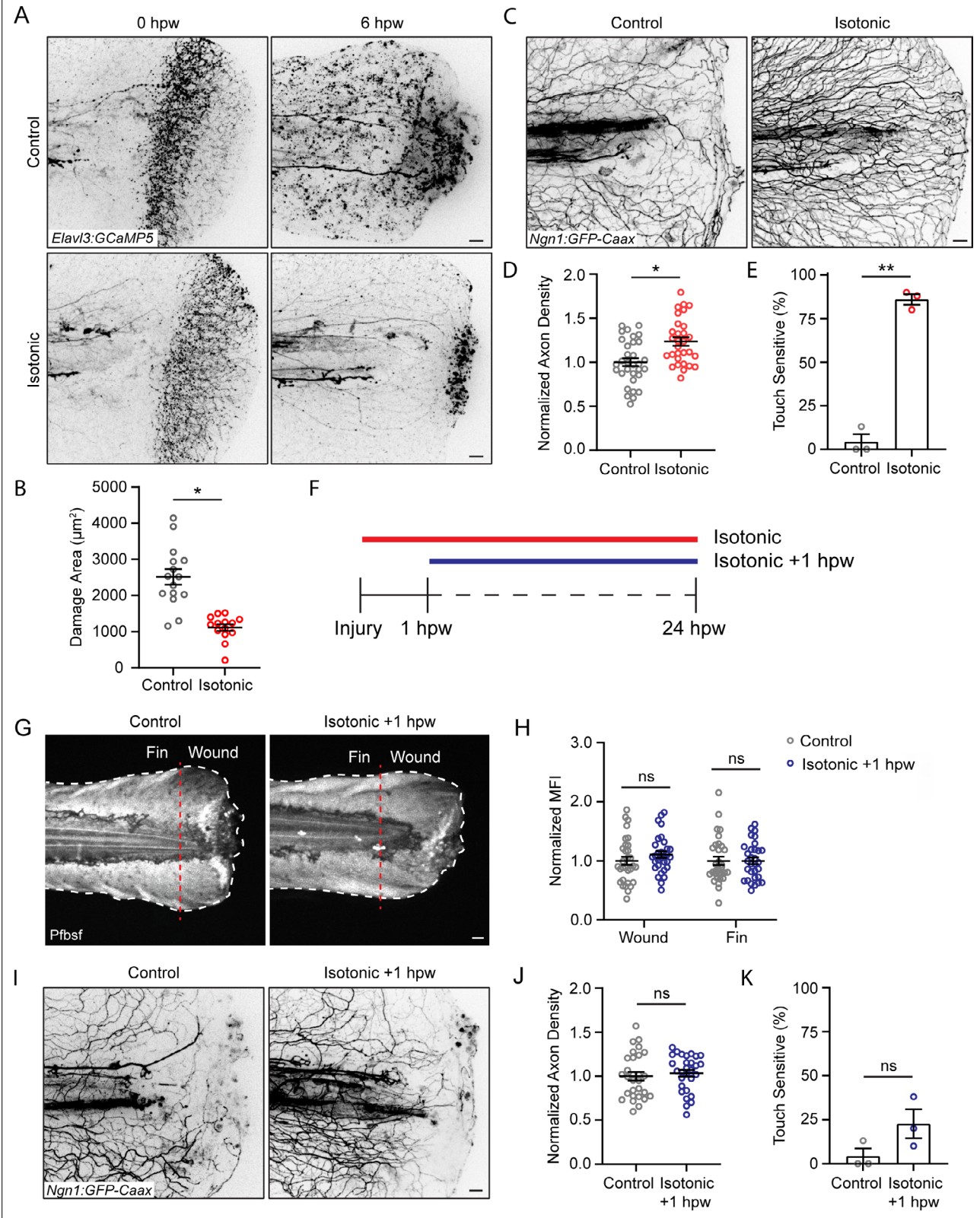

**Figure 6.** Isotonic treatment improves axon regeneration. (**A**) Confocal max-projected images of axon damage in control or isotonic-treated *Tg(Elavl3:GCaMP5)* larvae 0 or 6 hr post-wound (hpw). (**B**) Quantification of axon damage in control and isotonic-treated burned fins at 6 hpw as treated in A. N=16 larvae per condition from three replicates. (**C**) Confocal max-projected images of sensory axons in larvae 24 hpw as treated in A. (**D**) Quantification of axon density 24 hpw in larvae treated as depicted in C. N>30 larvae per condition from three replicates. (**E**) Quantification of

*Figure 6 continued on next page*

*Figure 6 continued*

sensory perception 24 hpw in larvae treated as in C. N=24 larvae each from three replicates. (**F**) Schematic illustrating the different isotonic treatment paradigms that are being compared. (**G**) Confocal sum-projected images of pentafluorobenzenesulfonyl fluorescein [Pfbsf] intensity in control and isotonic +1 hpw treated burned larvae. Dashed red line denotes the boundary between the wound area and distal fin tissue. White dashed line denotes the fin. (**H**) Quantification of mean Pfbsf fluorescence intensity (MFI) 6 hpw in the indicated region of the fin normalized to the control condition. N=31 larvae per condition from three replicates. (**I**) Confocal max-projected images of sensory axons 24 hpw in burned control or isotonic-treated larvae starting 1 hpw. (**J**) Quantification of axon density 24 hpw in larvae treated as in D. N=29 larvae per condition from three replicates. (**K**) Quantification of sensory perception 24 hpw in larvae treated as in D. N=24 larvae per condition from three replicates. Unless otherwise indicated, scale bars = 20 μm. *p<0.05, **p<0.01, ns = not significant.

The online version of this article includes the following source data and figure supplement(s) for figure 6:

**Source data 1.** Numerical data for *Figure 6B*.

**Source data 2.** Numerical data for *Figure 6D*.

**Source data 3.** Numerical data for *Figure 6E*.

**Source data 4.** Numerical data for *Figure 6H*.

**Source data 5.** Numerical data for *Figure 6J*.

**Source data 6.** Numerical data for *Figure 6K*.

**Figure supplement 1.** Keratinocyte movement after injury and effect of D-Sorbitol on sensory axon regeneration.

**Figure supplement 1—source data 1.** Numerical data for *Figure 6—figure supplement 1B*.

**Figure supplement 1—source data 2.** Numerical data for *Figure 6—figure supplement 1C*.

the spatial localization of signaling. Keratinocyte migration requires transiently detaching from neighboring cells. Thus, the act of migrating induces a physical change in the tissue that demarcates the wound region from healthy tissue. It is known that production of ROS promotes keratinocyte motility, and that adhesion is linked to cellular redox state (*Dunnill et al., 2017*; *Mendieta-Serrano et al., 2019*). Given these observations, it seems plausible that ROS production in wounded tissue is linked to the biomechanical state of keratinocytes – with low ROS in static, adhered cells, and high ROS in loosely adhered or migrating cells. A conceptual framework such as this would explain excessive ROS production in burn wounded tissue. Early keratinocyte dynamics in burned tissue are associated with normal wound edge ROS production. However, lack of a migratory stop signal may result in excessive keratinocyte migration and subsequent epithelial damage associated with keratinocytes detaching from the basal lamina. Therefore, failure to restore epithelial homeostasis due to unabated keratinocyte movement may allow for ROS production to continue over time and spread further away from the wound site. Future studies will be aimed at identifying the molecular link between cell migration and tissue scale signaling during tissue repair.

In summary, we have identified early wound-induced keratinocyte migration as a mechanism that controls spatial patterning of long-term wound signaling. These findings highlight the ability of keratinocytes within the wound microenvironment to integrate early signaling and migratory functions that mediate initial wound closure and subsequently regulate spatial tissue signaling necessary for efficient repair of sensory neuron function. Further, our results not only highlight the utility of larval zebrafish for revealing new insights of the tissue response to injury in vivo, but also demonstrate the potential for these findings to inform new treatment strategies for wound healing more broadly.

# Materials and methods

**Key resources table**

| Reagent type (species) or resource | Designation | Source or reference | Identifiers | Additional information |
|---|---|---|---|---|
| Strain background (*D. rerio*) | *WT (AB)* | ZIRC | ZL1 | https://zebrafish.org/home/guide.php |

*Continued on next page*

*Continued*

| Reagent type (species) or resource | Designation | Source or reference | Identifiers | Additional information |
|---|---|---|---|---|
| Strain background (*D. rerio*) | *Tg(Ngn1:GFP-Caax)* | *Blader et al., 2003* | | |
| Strain background (*D. rerio*) | *Tg(Krt4:LifeAct-mRuby)* | *Lam et al., 2015* | | |
| Strain background (*D. rerio*) | *Tg(Krt4:UtrCH-GFP)* | *Lam et al., 2015* | | |
| Strain background (*D. rerio*) | *Tg(Krt4:TdTomato)* | Huttenlocher lab | | |
| Strain background (*D. rerio*) | *TgBac(Lamc1:Lamc1-sfGFP)* | *Yamaguchi et al., 2022* | | |
| Strain background (*D. rerio*) | *Tg(Elavl3:GCaMP5)* | *Akerboom et al., 2012* | | Received from Jan Huisken lab |
| Strain background (*D. rerio*) | *Tg(Krtt1c19e:LifeAct-mRuby)* | This paper | | |
| Strain background (*D. rerio*) | *Tg(Krtt1c19e:acGFP)* | *Lee et al., 2014* | | Received from Alvaro Sagasti lab |
| Chemical compound, drug | FM 1-43 dye | Life Technologies | | |
| Chemical compound, drug | NaCl | Fisher Scientific | CAS 7647-14-5 | |
| Chemical compound, drug | D-Sorbitol | Sigma-Aldrich | CAS 50-70-4 | |
| Chemical compound, drug | CK666 (Arp2/3 inhibitor) | Sigma-Aldrich | CAS 442633-00-3 | |
| Chemical compound, drug | Sodium Azide | Fisher Scientific | CAS 26628-22-8 | |
| Chemical compound, drug | Pentafluorobenzenesulfonyl fluorescein | Santa Cruz | CAS 728912-45-6 | |
| Other | Cautery pen, fine tip | Bovie | AA01 | https://www.delasco.com/geiger/ |
| Other | Surgical blade No. 10 | Feather | 2976 | |
| Software, algorithm | GraphPad Prism | | RRID:SCR_002798 | https://www.graphpad.com/scientific-software/prism/ |
| Software, algorithm | Fuji, ImageJ | *Schneider et al., 2012* | RRID:SCR_002285 | https://fiji.sc/ |

## Zebrafish maintenance and handling

Adult zebrafish and embryos were maintained as described previously (*Houseright et al., 2021*; *Miskolci et al., 2019*). For all experiments, 3 dpf larvae were anesthetized in E3 medium containing 0.2 mg/mL Tricaine (ethyl 3-aminobenzoate; Sigma-Aldrich) and maintained at 28.5°C. All transgenic lines including *Tg(Ngn1:GFP-Caax)* (*Blader et al., 2003*), *Tg(Krt4:LifeAct-mRuby)* (*Lam et al., 2015*), *Tg(Krt4:UtrCH-GFP)*, *Tg(Krt4:TdTomato)*, *TgBac(Lamc1:Lamc1-sfGFP)* (*Yamaguchi et al., 2022*), *Tg(Elavl3:GCaMP5)* (*Akerboom et al., 2012*), *Tg(Krtt1c19e:LifeAct-mRuby)*, and *Tg(Krtt1c19e:acGFP)* were maintained on the AB background strain. To screen larvae for fluorescence, a Zeiss Zoomscope EMS3/SyCoP3 with a Plan-NeoFluar Z objective was used.

## Generation of *Tg(Krtt1c19e:LifeAct-mRuby)* transgenic line

The *Krtt1c19e* promoter (*Lee et al., 2014*) flanked by Age1 and NotI was isolated and cloned into an expression vector containing Lifeact-mRuby and Tol2 elements for genomic integration. 3 nL of solution made of 50 ng DNA and 25 ng Tol2 transposase mRNA were injected into the yolk of one-cell

stage embryos. F0 larvae were raised to adulthood and crossed to adult AB zebrafish. F2 larvae were screened for mRuby expression and grown to generate stable lines. *Tg(Ngn1:GFP-Caax)* larvae transiently expressing *Krtt1c19e:Lifeact-m*Ruby were used for simultaneous imaging of axon-keratinocyte interactions, and larvae transiently expressing *Krtt1c19e:Lifeact-mRuby* were used to acquire time-lapses of basal keratinocyte migration in the CK666 treatment condition.

## Caudal fin transection and burn injury

All injuries were applied to fish anesthetized in 1× Tricaine with E3. Transection of the caudal fin was performed on anesthetized larvae in a 60 mm tissue culture-treated dish containing 1× Tricaine with E3. Larvae were cut perpendicular to the caudal notochord boundary using a surgical blade (Feather No. 10, VWR). Burn injury was performed on anesthetized larvae in a 60 mm tissue culture-treated dish containing 1× Tricaine with E3. A fine tip cautery pen (Geiger Medical Technologies) was used to burn the caudal fin until the wounded region reached halfway to the posterior notochord boundary. After injury, larvae were kept in 60 mm dishes and maintained at 28.5°C until imaged. For two-wound experiments, larvae were either transected or burned as described above. Secondary transection, after either 5 min or 6 hr, was performed as described above.

## Drug treatment

For all treatments, larvae were incubated in the indicated drug solution for at least 15–30 min. Each drug solution was made containing 1× Tricaine with E3 to keep larvae anesthetized during the experiment. Unless indicated otherwise, larvae were in the presence of treatment for the duration of all experiments. All treatments did not obviously impair larval development or health, and axon density in unwounded larvae was measured for each treatment to ensure there were no deficits in axonal patterning (*Appendix 1—figure 1*). To elicit sensory axon damage in the absence of a wound, larvae were treated with 1.5% sodium azide (Fisher Scientific). Isotonic medium was prepared by supplementing 1× Tricaine with E3 with either NaCl (Fisher Scientific) or D-Sorbitol (Sigma-Aldrich) to a final concentration of 135 mM. Isotonic medium does not noticeably impair embryonic development or health of the larvae on the time scale used here (*Gault et al., 2014*; *Jelcic et al., 2019*). For experiments using CK666 (Sigma), larvae were incubated in 100 µM CK666 for 1 hr before wounding and kept in treatment until 6 hpw. The DPI dosage of 20 µM used in this manuscript was determined by dosage curve of 1–200 µM. Larvae were pre-treated for 1 hr and then burned and incubated in treatment for 1 hr – treatments over 20 µM or over 1 hpw were detrimental to larval health and development.

## Live and time-lapse imaging

Larvae were imaged using a spinning disc microscope (CSU-X, Yokogawa) with a confocal scanhead on a Zeiss Observer Z.1 inverted microscope, Plan-Apochromat NA 0.8/20× objective, and a Photometrics Evolve EMCCD camera. All images were acquired using ZEN 2.6 software. For time-lapse imaging, larvae were mounted in a zWEDGI restraining device (*Huemer et al., 2017*) with the head covered in 2% low-melting point agarose (Sigma-Aldrich). For single time point imaging, anesthetized larvae were mounted in 2% low-melting point agarose on a 35 mm glass-bottom depression dish (CellVis). In all cases, larvae were imaged in E3 medium supplemented with Tricaine as described above.

## Quantification of axon density and sensory function

Axon density was measured by generating maximum intensity z-projected confocal images of the caudal fin using Fiji (*Schneider et al., 2012*). For every experiment, the caudal fin area posterior to the notochord was outlined using the Polygon tool and measured to obtain a total surface area ROI. Axons inside the outlined area were manually thresholded so all axons posterior to the notochord were labeled and no saturated pixels were present. Density was measured by dividing the area of detected axons by the area of the ROI. In each case, density values of the experimental sample were normalized to the indicated control – either unwounded or control-treated fins. Sensory neuron function was determined using a behavioral touch assay (*Granato et al., 1996*). 3 dpf larvae were wounded as described above. At the indicated time post-wound, larvae were briefly anesthetized for mounting into the zWEDGI restraining device, with only the head mounted in 2% low-melting point agarose. Fresh E3 was added, and larvae were allowed to rest for 1 hr. To assess sensory function, the

wounded region of caudal fin was touched with the tip of an eyelash brush (No. 1 Superfine Eyelash, Ted Pella) and the presence or absence of a twitch reflex was recorded.

## Quantification of axon damage

Maximum intensity z-projected confocal images of the caudal fin were generated using Fiji. For all experiments, the caudal fin area posterior to the notochord was outlined using the Polygon tool and measured to obtain a total surface area ROI. Axon fragments inside the outlined area were manually thresholded so all fragments posterior to the notochord were labeled and no saturated pixels were present, and an area measurement of these thresholded pixels was taken.

## Visualization of sensory axon and tissue damage

To visualize damage to sensory axons, *Tg(Elavl3:GCaMP5)* larvae were used. Identical microscope settings (20× objective, 10% laser, 100 ms exposure, 2 μm step size) were used for all experiments to acquire images and movies. Representative images are maximum intensity z-projections of the caudal fin generated using Fiji. FM 1–43 dye (Life Technologies) was used to visualize tissue damage following transection and burn injury. For these experiments, larvae were incubated in 1 mg/mL FM 1–43 for 15 min prior to injury and through time at which they were imaged. Larvae were maintained at 28.5°C until imaging at the indicated time post-injury.

## Quantification of hydrogen peroxide level

Hydrogen peroxide was quantified using Pfbsf (Santa Cruz) (*Maeda et al., 2004*). Larvae were incubated in 1 μM Pfbsf for 15 min prior to injury and maintained in dye solution for the duration of each experiment. Identical microscope settings (10× objective, 1% laser, 50 ms exposure, 3.7 μm step size) were used for all experiments to acquire images and movies. Pfbsf intensity was calculated by generating sum projections and measuring mean gray value of the fin and wound zone in Fiji. Wound zone Pfbsf quantifications were taken by measuring mean gray value of the area posterior to the notochord. For measurements of the fin, mean gray value of the trunk area 200 μm anterior to the tip of the notochord and excluding pigmented skin within the region was measured. Background signal was subtracted for each measurement.

## Cell tracking

Basal keratinocyte tracking following tissue injury was performed using *Tg(Krtt1c19e:acGFP)* larvae. Cell tracking was performed using the Spots module in Imaris version 9.8.2 (Bitplane, Zurich, Switzerland). For each larva, three representative cells were identified and manual tracking was performed, with the average of these cells being used to generate a single value for further analysis. To control for drift of the entire fin during imaging, non-moving pigment was manually tracked by Brightfield and track length was subtracted from basal keratinocyte movement. In all cases, larvae were imaged for 1 hr following injury at an interval of 30 s.

## Morpholino injection

Ngn1 morpholino with the sequence 5'-ACG ATC TCC ATT GTT GAT AAC CTG G-3' (*Cornell and Eisen, 2002*) was used to prevent sensory neuron formation in Elavl3-GCaMP5 larvae to confirm damage signals were constrained only to axons. 5 ng of Ngn1 morpholino was injected into the yolk of one- to two-cell stage zebrafish embryos. Larvae were incubated at 28.5°C until used for experiments at 3 dpf. Before use in experiments, larvae were screened by the sensory function assay described above to ensure that sensory neurons were depleted.

## Image processing

Images were processed and analyzed using Fiji and Imaris version 9.8.2 as indicated. Supplemental movies were generated in Fiji and edited using Adobe Premiere Pro (Adobe). In Adobe Premiere Pro, pseudocoloring of individual keratinocytes was done using the Color Effects module with manual tracking.

## Statistical analysis

Each experimental condition consists of at least three independent biological replicates, defined as three clutches of larvae spawned on 3 different days. Cell-tracking experiments were analyzed

using non-parametric methods (Wilcoxon rank-sum test). Quantification of axon density was analyzed using linear mixed-effect models in which biological replicate was treated as a random effect and experimental conditions (e.g. wound, time, or chemical treatment) treated as fixed factors. Experiments measuring fluorescence intensity (Pfbsf intensity) were analyzed in the same manner, except the response (fluorescence) was log-transformed prior to analysis. Means computed on the log scale estimate the median response when back-transformed to original units. At the same time, differences between means (of log-transformed data) become ratios of medians after back transformation to the original scale (*Aitkin et al., 1989*). Experiments involving the proportion of fish that responded to touch were analyzed using a general linear model that included replicate and experimental condition as fixed effects; standard errors used for estimation and testing were adjusted to correct for heteroscedasticity in the proportions (*Long and Ervin, 2000*). Graphing was performed using GraphPad Prism 9 (GraphPad Software, Inc, San Diego, CA, USA). Sample size is reported for specific experiments in the Figure legends.

## Acknowledgements

We would like to thank Dr. Mary Halloran (University of Wisconsin-Madison) for the gift of the *Tg(Ngn1- GFP-Caax)* fish line, Dr. Jan Huisken (University of Göttingen) for the *Tg(Elavl3-GcaMP5)* fish line, Dr. Alvaro Sagasti (University of California Los Angeles) for the *Tg(Krtt1c19e:acGFP)* fish line, and Dr. Holger Knaut (New York University) for the *TgBac(LamC1:LamC1-sfGFP)* fish line. We would like to thank Taylor Schoen and Veronika Miskolci for their critical reading of the manuscript, and the members of the Huttenlocher lab for their thoughtful input throughout this project. The authors acknowledge K99 GM147303 to Adam Horn and R35 GM118027 to Anna Huttenlocher.

## Additional information

### Funding

| Funder | Grant reference number | Author |
| --- | --- | --- |
| National Institutes of Health | K99 GM147303 | Adam Horn |
| National Institutes of Health | R35 GM118027 | Anna Huttenlocher |

The funders had no role in study design, data collection and interpretation, or the decision to submit the work for publication.

### Author contributions

Alexandra M Fister, Adam Horn, Conceptualization, Data curation, Formal analysis, Validation, Investigation, Visualization, Methodology, Writing - original draft, Writing - review and editing; Michael R Lasarev, Formal analysis, Writing - review and editing; Anna Huttenlocher, Conceptualization, Supervision, Funding acquisition, Methodology, Writing - original draft, Project administration, Writing - review and editing

### Author ORCIDs

Adam Horn ⬢ http://orcid.org/0000-0002-2802-3621
Michael R Lasarev ⬢ https://orcid.org/0000-0002-1896-2705
Anna Huttenlocher ⬢ https://orcid.org/0000-0001-7940-6254

### Ethics

This study was carried out in accordance with the recommendations from the Guide for the Care and Use of Laboratory Animals from the National Institutes of Health. All zebrafish protocols in this study were approved by the University of Wisconsin-Madison Research Animals Resource Center (Protocol M005405-R02).

Reviewer #1 (Public review): https://doi.org/10.7554/eLife.94995.3.sa1

Reviewer #3 (Public review): https://doi.org/10.7554/eLife.94995.3.sa2
Author response https://doi.org/10.7554/eLife.94995.3.sa3

## Additional files

### Supplementary files
- MDAR checklist
- Source data 1. Primer source data.

### Data availability
All data generated or analyzed during this study are included in the manuscript and source files.

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

# Appendix 1

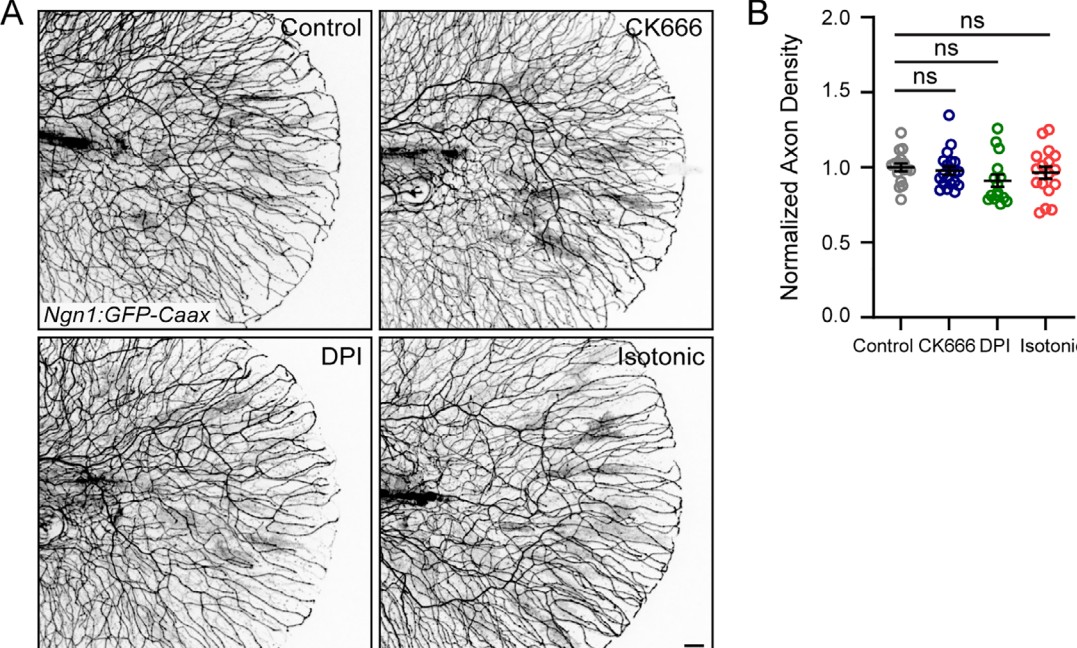

**Appendix 1—figure 1.** Drug treatments do not affect axon density in unwounded larvae. (**A**) Representative max-projected confocal images of sensory axons in unwounded, drug-treated 4 dpf larvae. (**B**) Quantification of axon density in larvae treated as stated in E. N>15 larvae each from 3 replicates. Scale bar = 20 μm. ns=not significant.

The online version of this article includes the following source data for appendix 1—figure 1:

**Appendix 1—figure 1—source data 1.** Numerical data for *Appendix 1—figure 1*.

**Appendix 1—figure 1—source data 2.** Numerical data for *Appendix 1—figure 1*.

