## [Editor Report · eLife assessment]

This **important** study identifies a novel link between the early keratinocyte response to wounds and the subsequent regenerative capacity of local sensory neurons. The evidence supporting the claims of the authors is **convincing**, although inclusion of conditional genetics or cell-autonomy tests would have strengthened the mechanistic aspects. The work will be of interest to cell and developmental biologists interested in tissue regeneration and cell interactions in a broader context.

---

## [Referee Report · Reviewer #1 (Public review)]

Summary:

In this manuscript, Fister et. al. investigate how amputational and burn wounds affect sensory axonal damage and regeneration in a zebrafish model system. The authors discovered that burn injury results in increased peripheral axon damage and impaired regeneration. Convincing experiments show altered axonal morphology and increased Ca2+ fluxes as a result of burn damage. Further experimental proof supports that early removal of the burnt tissue by amputation rescues axonal damage. Burn damage was also shown to markedly increase keratinocyte migration and increase localized ROS production as measured by the dye Pfbsf. These responses could be inhibited by Arp 2/3 inhibition and isotonic treatment.

Strengths:

The authors use state-of-the-art methods to study and compare transection and burn-induced tissue damage. Multiple experimental approaches (morphology, Ca2+ fluxing, cell membrane labeling) confirm axonal damage and the impaired regeneration time. Furthermore, the results are also accompanied by functional response tests of touch sensitivity. This is the first study to extend the role of tissue-damage related osmotic exposure beyond wound closure and leukocyte migration to a novel layer of pathology: axonal damage and regeneration.

The authors provide elegant experiments showing that early removal of the burnt tissue can rescue damage-induced axonal damage, which could also be interpreted in an osmotic manner. In the revised version of the paper the authors indeed show that tail fin transections close faster than burn wounds, allowing for lower hypotonic exposure time. However, their new experiments suggest that axonal damage and slow regeneration in tail fin burn wounds are not a direct consequence of the extended exposure time to hypotonic water.

Weaknesses:

The conclusions of the paper claiming a link between burn-induced epithelial cell migration, spatial redox signaling, and sensory axon regeneration are mainly based on correlative observations. Arp 2/3 inhibition impairs cell migration but has no significant effect on axon regeneration and restoration of touch sensitivity.

Genetic approaches have been tested during the revision process to directly prove the role of ROS production by targeting DUOX, however, the combination of DUOX morpholino and burn injury was lethal to the larvae and long-term pharmacological inhibition over 1 hour was also detrimental.

---

## [Referee Report · Reviewer #3 (Public review)]

Fister and colleagues use regeneration of the larval zebrafish caudal fin to compare the effects of two modes of tissue damage-transection and burn-on cutaneous sensory axon regeneration. The authors found that restoration of sensory axon density and function is delayed following burn injury compared to transection.

The authors hypothesized that thermal injury triggers signals within the wound microenvironment that impair sensory neuron regeneration. The authors identify differences in the responses of epithelial keratinocytes to the two modes of injury: keratinocytes migrate in response to burn but not transection. Inhibiting keratinocyte migration with a small-molecule inhibitor of Arp2/3 (CK666) resulted in decreased production of reactive oxygen species (ROS) at early, but not late, timepoints. Preventing keratinocyte migration by wounding in isotonic media resulted in increased sensory function 24 hours after burn.

Strengths of the study include the beautiful imaging and rigorous statistical approaches used by the authors. The ability to assess both axon density and axon function during regeneration is quite powerful. The touch assay adds a unique component to the paper and strengthens the argument that burns are more damaging to sensory structures and that different treatments help to ameliorate this.

A weakness of the study is the lack of genetic and cell autonomous manipulations. Additional comparisons between transection and burns, in particular with manipulations that specifically modulate ROS generation or cell migration without potentially confounding effects on other cell types or processes would help to strengthen the manuscript. In terms of framing their results, the authors refer to "sensory neurons" and "sensory axons" throughout the text - it should be made clear what type of neuron(s)/axon(s) are being visualized/assayed. Along these lines, a broader discussion of how burn injuries affect sensory function in other systems-and how the authors' results might inform our understanding of these injury responses-would be beneficial to the reader.

In summary, the authors have established a tractable vertebrate system to investigate different sensory axon wound healing outcomes in vivo that may ultimately allow for the identification of improved treatment strategies for human burn patients. Although the study implicates differences in keratinocyte migration and associated ROS production in sensory axon wound healing outcomes, the links between these processes could be more rigorously established.

---

## [Author Response]

The following is the authors’ response to the original reviews.

**Reviewer #1 (Public Review):**
Summary:In this manuscript, Fister et. al. investigate how amputational and burn wounds affect sensory axonal damage and regeneration in a zebrafish model system. The authors discovered that burn injury results in increased peripheral axon damage and impaired regeneration. Convincing experiments show altered axonal morphology and increased Ca2+ fluxes as a result of burn damage. Further experimental proof supports that early removal of the burnt tissue by amputation rescues axonal damage. Burn damage was also shown to markedly increase keratinocyte migration and increase localized ROS production as measured by the dye Pfbsf. These responses could be inhibited by Arp 2/3 inhibition and isotonic treatment.Strengths:The authors use state-of-the-art methods to study and compare transection and burn-induced tissue damage. Multiple experimental approaches (morphology, Ca2+ fluxing, cell membrane labeling) confirm axonal damage and impaired regeneration time. Furthermore, the results are also accompanied by functional response tests of touch sensitivity. This is the first study to extend the role of tissue-damage-related osmotic exposure beyond wound closure and leukocyte migration to a novel layer of pathology: axonal damage and regeneration.Weaknesses:The conclusions of the paper claiming a link between burn-induced epithelial cell migration, spatial redox signaling, and sensory axon regeneration are mainly based on correlative observations. Arp 2/3 inhibition impairs cell migration but has no significant effect on axon regeneration and restoration of touch sensitivity.

We agree with the reviewer. We have tried many experiments to address this question. The data show that Arp 2/3 inhibition with CK666 is an effective way to inhibit initial keratinocyte migration. However, later migration still proceeds. What is interesting is that just inhibition of the early migration is sufficient to restore localized ROS production in the wound area in the first hour post-burn, even if this is not sufficient to prevent ROS accumulation over time. There is also a trend toward improved sensory neuron function late after this early treatment. However, this is not statistically significant. We think it is likely that both migration and tissue scale ROS influence the regeneration defect of sensory neurons after burn. The data using isotonic solution supports this conclusion. We have tried many other ways to limit keratinocyte migration including depletion of talin and expression of a dominant negative Rac in basal epithelial cells, but these treatments were not compatible with survival of the fish after burn.

Pharmacological or genetic approaches should be used to prove the role of ROS production by directly targeting the known H2O2 source in the system: DUOX.

We agree that pharmacologic or genetic approaches to directly manipulate ROS production would provide substantial support to the hypothesis that ROS, along with keratinocyte migration, is a main factor contributing to poor burn outcomes. To address this, we first tried using a morpholino to deplete DUOX. However, the combination of DUOX morpholino and burn injury was lethal to larvae. We also used pharmacologic inhibition of ROS production using DPI (Diphenyleneiodonium). With this treatment, ROS is inhibited for only the first hour post-burn as treatment is lethal for longer periods of time. Burned larvae have marginally improved axon density and touch sensitivity, suggesting the importance of ROS in burn outcomes, however it was not statistically significant. It is likely that an increased effect would be observed with longer treatment, but treatment for more than 1 hour was toxic. We have added a supplemental figure with this new DPI data.

While the authors provide clear and compelling proof that osmotic responses lie at the heart of the burn-induced axonal damage responses, they did not consider the option of further exploring any biology related to osmotic cell swelling. Could osmotic ATP release maybe play a role through excitotoxicity? Could cPLA2 activation-dependent eicosanoid production relate to the process? Pharmacological tests using purinergic receptor inhibition or blockage of eicosanoid production could answer these questions.

We agree that the role of osmotic cell swelling in the burn response is an interesting avenue for future study. However, we make use of isotonic treatment in this study specifically for its effect on keratinocyte migration and broad-scale wound healing. As a result, we feel that pursuing the biology of this swelling phenomenon is outside the scope of this paper.

The authors provide elegant experiments showing that early removal of the burnt tissue can rescue damage-induced axonal damage, which could also be interpreted in an osmotic manner: tail fin transections could close faster than burn wounds, allowing for lower hypotonic exposure time. Axonal damage and slow regeneration in tail fin burn wounds could be a direct consequence of extended exposure time to hypotonic water.

We have done experiments using FM dye to test how long it takes burn and transection wounds to close (shown below). In these experiments, dye entry into wounded tissue is used as a readout of wound closure. Dye is only able to enter wounded tissue when the epithelial barrier is disrupted. Our data reveal that transections take approximately 10 minutes to fully close, while burns take approximately 20 minutes to close.

To test if this difference in wound closure time would have an effect on axon outcomes, we repeated, but slightly modified, the dual-wound experiment. We increased the amount of time the burn condition was exposed to hypotonic conditions by 10 additional minutes (by transecting burned tissue at 15 minutes post burn, shortly before closure) and compared axon outcomes to the 5 mpw control transection. These results show there was no difference in axon regeneration or function when secondary transection was performed at 5 or 15 minutes post burn, suggesting that increased exposure to hypotonic solution is not the reason for defects in axon outcomes after burn injury.

**Author response image 2. sa3fig2:** 

**Reviewer #2 (Public Review):**
This is an interesting study in which the authors show that a thermal injury leads to extensive sensory axon damage and impaired regrowth compared to a mechanical transection injury. This correlates with increased keratinocyte migration. That migration is inhibited by CK666 drug treatment and isotonic medium. Both restrict ROS signalling to the wound edge. In addition, the isotonic medium also rescues the regrowth of sensory axons and recovery of sensory function. The findings may have implications for understanding non-optimal re-innervation of burn wounds in mammals.The interpretation of results is generally cautious and controls are robust.Here are some suggestions for additional discussion:The study compares burn injury which produces a diffuse injury to a mechanical cut injury which produces focal damage. It would help the reader to give a definition of wound edge in the burn situation. Is the thermally injured tissue completely dead and is resorbed or do axons have to grow into damaged tissue? The two-cut model suggests the latter. Also giving timescales would help, e.g. when do axons grow in relation to keratinocyte movement? An introductory cartoon might help.

We thank the reviewer for these insightful comments and questions. The burn wound is defined as the area that is directly damaged as a result of increased heat (labeled by FM dye entry), and the burn wound edge as the first line of healthy cells adjacent to the burned cells. These definitions have been added to the text to clarify the areas referenced. Recent experiments lead us to believe the wound area is composed almost completely of dead cells, but we are currently working to discover the fate of these dead cells as well as the wound adjacent cells that migrate to the wound edge after burn. As a result, we do not know whether axons grow into damaged tissue or if the damaged tissue is extruded, but we do see growth cone formation within a few hours after wounding suggesting the axons are actively trying to regenerate after a burn.

Could treatment with CK666 or isotonic solution influence sensory axons directly, or through other non-keratinocyte cell types, such as immune cells?

We have done experiments looking at the density of caudal fin innervation in CK666, isotonic, or DPI treated fins. The axon density is unchanged in all these treatments compared to control treated larvae, so we do not believe these treatments affect axon health homeostatically. These data have been added to supplemental figure 3. Additionally, one of the benefits of the larval zebrafish burn model is the simplicity of the system – the epidermis is primarily composed of sensory axons, mesenchymal cells and keratinocytes. The burn environment is proinflammatory so it does promote immune cell recruitment, but we do not believe the immune cells are interacting directly with sensory axons besides clearing axonal debris. Previous papers by our lab have shown that peak immune cell recruitment occurs at 6 hpw, but they localize to the damaged tissue in the burn area and not the wound edge.

**Reviewer #3 (Public Review):**
Fister and colleagues use regeneration of the larval zebrafish caudal fin to compare the effects of two modes of tissue damage-transection and burn-on cutaneous sensory axon regeneration. The authors found that restoration of sensory axon density and function is delayed following burn injury compared to transection.The authors hypothesized that thermal injury triggers signals within the wound microenvironment that impair sensory neuron regeneration. The authors identify differences in the responses of epithelial keratinocytes to the two modes of injury: keratinocytes migrate in response to burn but not transection. Inhibiting keratinocyte migration with the small-molecule inhibitor of Arp2/3 (CK666) resulted in decreased production of reactive oxygen species (ROS) at early, but not late, time points. Preventing keratinocyte migration by wounding in isotonic media resulted in increased sensory function 24 hours after burn.Strengths of the study include the beautiful imaging and rigorous statistical approaches used by the authors. The ability to assess both axon density and axon function during regeneration is quite powerful. The touch assay adds a unique component to the paper and strengthens the argument that burns are more damaging to sensory structures and that different treatments help to ameliorate this.A weakness of the study is the lack of genetic and cell-autonomous manipulations. Additional comparisons between transection and burns, in particular with manipulations that specifically modulate ROS generation or cell migration without potentially confounding effects on other cell types or processes would help to strengthen the manuscript.

The use of genetic and cell-autonomous approaches would strengthen our study, however, we were unable to do this due to the lethality of these genetic approaches (or cell autonomous approaches). Basal epithelial migration is necessary for embryonic development. We attempted to circumvent this by generation of larvae transiently expressing a dominant-negative form of Rac, a protein crucial to the migratory process. The chimeric expression of the dominant negative Rac was either damaging to the larvae or the mosaicism was too low to observe any effects on migration phenotype.

We also attempted a genetic approach to manipulate ROS production, as discussed above. We found that the DUOX morpholino was lethal to burned larvae. Finally, we attempted pharmacological inhibition of ROS production using the inhibitor DPI (Diphenyleneiodonium). With this treatment, burned larvae have marginally improved axon density and touch sensitivity, suggesting that dampening ROS may improve outcome. The DPI data have been added to the manuscript.

In terms of framing their results, the authors refer to "sensory neurons" and "sensory axons" throughout the text - it should be made clear what type of neuron(s)/axon(s) are being visualized/assayed. Along these lines, a broader discussion of how burn injuries affect sensory function in other systems - and how the authors' results might inform our understanding of these injury responses - would be beneficial to the reader.In summary, the authors have established a tractable vertebrate system to investigate different sensory axon wound healing outcomes in vivo that may ultimately allow for the identification of improved treatment strategies for human burn patients. Although the study implicates differences in keratinocyte migration and associated ROS production in sensory axon wound healing outcomes, the links between these processes could be more rigorously established.

The inconsistency between “neuron” and “axon” has been noted and the text has been corrected accordingly. “Neuron” is used when referring to the cell as a whole, while “axon” is used when referring to the sensory processes in the caudal fin. We added information about burn in the introduction as suggested: “While epithelial tissue is well adapted to repair from mechanical damage, burn wounds heal poorly. Thermal injury results in chronic pain and lack of sensation in the affected tissue, suggesting that an abnormal sensory neuron response contributes to burn wound pathophysiology.”

We thank the reviewer’s for their comments.

**Recommendations For The Authors:**

**Reviewer #1 (Recommendations For The Authors):**
Suggested experiments:(1) ROS measurements with the dye Pfbsf should be validated with more established ROS probes such as HyPer.

Pfbsf has been used previously as a readout of ROS production, and its use is documented in zebrafish (Maeda et al., *Angew Chem Int Ed Engl*, 2004, and Niethammer et al, *Nature*, 2009). These sources have been added as references when introducing Pfbsf to provide context for its use. The probe was validated and compared to HyPer in Niethammer’s 2009 paper. In our hands, we have used both probes and have similar results with tail transection.

(2) To better support claims on ROS and H2O2 playing a central role in mediating axonal damage, the authors should consider pharmacological approaches such as rescue experiments with H2O2 and experiments using inhibitors such as DPI ar apocynin.While the above reagents and drugs have limitations and non-specific side effects, more convincing proof could result from genetic approaches including experiments on DOUX knockdown or knockout lines.

To further dissect the role of ROS in the burn response, we conducted experiments using DPI, a potent ROS inhibitor that is well-documented in the literature. We found that 20 uM treatment of DPI (1 hour pretreatment, 1 hour post-burn) marginally improved axon density when quantified 24 hpw. Any higher dose, when in combination with a burn, proved to be lethal. Longer treatment with DPI was also not tolerated.

In addition to experiments with DPI, we attempted to burn larvae that were injected with DUOX morpholino. The combined use of burn and DUOX MO was lethal. We have dampened the conclusions and include the new data with the DPI in the revised manuscript.

Minor corrections:(1)A phrase/expression in the abstract is confusing: isotonic treatment does not "induce osmotic regulation". Cells exposed to hypo- or hypertonicity will respond by regulatory volume decrease or increase, respectively. Isotonic treatment maintains homeostasis.

We appreciate this point and agree with the distinction. Revisions have been made in the text accordingly.

(2) Figures 4E and 5E would be better to show as an average of multiple experiments with statistical significance.

The purpose of figures 4E and 5E are to demonstrate changes in fluorescence intensity and localization of ROS using the representative time series shown in 4D and 5D. The figure legend has been updated accordingly.

**Reviewer #2 (Recommendations For The Authors):**
Figure 3D How can one distinguish between the two cellular elements that randomly meet or that there is actual coordination? Can the interactions be quantified? It is also unclear what the authors mean by "sensory neuron movement". The authors show that the neuronal cell bodies stay in their position, so only the axons change position. Do they do this by growth, i.e. the neuronal growth cones follow the keratinocytes or do keratinocytes displace the axon shafts?

We have included supplemental movies that address this question in the new uploaded document. Figure 3D is comprised of still images taken from supplemental movie 2, which is a timelapse of keratinocytes/axons moving together after a burn injury. This movie clearly shows keratinocytes and their ensheathed axons moving simultaneously, so keratinocytes are mechanically pulling sensory axon shafts with them. We have revised the text to say axon movement, not sensory neuron movement.

Over the time course of axonal movement (1 hour post-burn), it is not possible that neuronal growth cones contribute to movement, as this is too slow – previous work by other labs has shown that it takes several hours for axons to fully regenerate into amputated tissue, with movement not even noticeable until about 3 hours post-wound (Rieger and Sagasti, PLOS Biology, 2011).

Regarding the second point, “neuron” vs. “axon” is an inconsistency in the text that has been corrected. “Neuron” is used when referring to the cell as a whole, “axon” is used when referring to the processes that innervate the caudal fin. The axons are physically pulled along with keratinocytes as they migrate after burn application. From our observations, growth cones appear closer to the wound site after the movement has stopped.

Figure 4G It is surprising that the visual differences in the distribution of values are not statistically significant.

The distribution of values in 4G was large and that is why there is no statistically-significant difference – we were also surprised at this result. We did all statistics with a statistician and this included rigorous criteria for significance.

Figure 4H The images seem to show a difference, whereas the quantification does not. I suggest choosing more representative images.

Figure 4H has been updated to include a more representative image of axon patterning with CK666 treatment.

Figure 6A The text states that axon damage in the control and isotonic condition is comparable, yet in the image, it appears that the damage in the isotonic treatment at 0 hpw is more distal.

This is a good observation that we consistently see in isotonic-treated fish after burn. Axon damage localizes more proximally in isotonic-treated samples because the keratinocytes distal to the notochord are likely dead, and the axons innervating those cells are likely immediately destroyed upon burn application. As a result, the distal axons are not present to express GCaMP. We believe isotonic treatment allows keratinocytes to live slightly longer, so axon damage is therefore prevented for longer. This is also the focus of continuing work to further understand the burn microenvironment.

Finally, the materials section could mention bias mitigation measures, e.g. withholding the treatment condition from the experimenter in the touch test.

We minimized bias in experiments whenever possible, and the conservative statistical measures that were applied to our data further reduce the likelihood of false significance.

**Reviewer #3 (Recommendations For The Authors):**
- Line numbers would have facilitated reviewer feedback.- Supplementary movies were missing in the submission.

The lack of supplementary movies upon submission was a mistake and the movies have been uploaded along with the revised manuscript.

Introduction:- Pg. 3: "In response to tissue damage, sensory neurons undergo rapid and localized axonal degeneration 4,5." Not sure reference 4 (Reyes et al) is appropriate here as this study was not in the context of tissue damage.

We have revised this section as suggested by the reviewer.

Results:- The expected expression pattern/localization of several transgenes was unclear. Please clearly state what cell type(s) each should label. For example, pg. 5 - "We next sought to further investigate sensory neuron function in burned tissue. For this, we assessed wound-induced axonal damage using zebrafish larvae that express the calcium probe GCaMP." Where is GCaMP expressed?

The manuscript has been updated to include expression patterns for the included transgenes – in this mentioned case, GCaMP is expressed in neurons under the pan-neuronal Elavl3 promoter.

- Introducing the GCaMP labeling could use some clarification. Pg. 5 - "As shown previously by other groups, GCaMP labels degenerating neurons in real time35." This is confusing. Do the authors mean that GCaMP increases immediately prior to Wallerian degeneration as shown by Vargas et al. (PMID: 26558774)?

Sustained elevated calcium levels are associated with axon damage. Previous work from other labs has shown that calcium influx follows axon injury (Ziv and Spira, EJN 1993, Adalbert et al., Neuroscience 2012). In these experiments, whenever there are CGaMP-positive punctae, this indicates axon damage. We have revised the manuscript to address this critique.

The Elavl3-GCaMP5 transgenic line will label when calcium levels increase in neurons. However, given the parameters used for imaging in our study (20x magnification, 100 ms exposure, and collection speed every 30 seconds for timelapses), we believe that only sufficiently large increases in calcium that are indicative of cell damage, and not physiological function, are being visualized.

- Figure 1E - Are these panels images of the same fish? Please specify in the legend.

Figure 1E is comprised of one transected and one burned larva each, live-imaged over the course of six hours. The legend has been updated to include this information.

- Figure 1F - How was the damage area measured? Consider doing this measurement over time to match Figure 1E.

Axon damage area measurements were performed similar to axon density measurements – maximum intensity z-projected confocal images of the caudal fin were generated using FIJI. For all experiments, the caudal fin area posterior to the notochord was outlined using the Polygon tool and measured to obtain a total surface area ROI. Axon fragments inside the outlined area were manually thresholded so all fragments posterior to the notochord were labeled and no saturated pixels were present, and an area measurement of these thresholded pixels was taken. We have added a section describing these measurements in the Methods section under “Axon damage quantification.”

- Pg. 5 - When introducing the ngn1 MO - please state the expected phenotype and cite the appropriate background literature_._

The ngn1 morpholino was cited in the Methods section with the appropriate literature (Cornell and Eisen, *Development*, 2002), from which we got the morpholino sequence. We thank the reviewer for pointing out the need for more introduction and clarification in the main text, so the ngn1 morpholino has been discussed in greater depth and cited in the main text as well using the same citation.

- The two-wound model is an elegant approach but could be more clearly described in the main text.

An improved explanation of the two-wound experiment has been added to the text.

- For Figure 3, it would be helpful to have a schematic of the anatomy illustrating the relative positions of axons and epidermal cell types.- Figure 3C - should an additional control here be transected? Given that the krt4:lifeact transgene labels both layers of the epidermis, how were the superficial and basal keratinocytes separated? Interpretation of this section should be carefully worded. The authors state that "...suggesting that the superficial keratinocytes are being pulled by the motile basal keratinocytes" (pg.7) but isn't another possibility that the superficial cells are stationary?

It is correct that the krt4:lifeact transgene labels both layers of keratinocytes, which together span 20-30 microns. These layers were separated from the same z-stack collected by confocal imaging. The first z-slice and last z-slice of the same stack were separated using FIJI and pseudocolored to appear as different colors. This clarification has been added to the Methods.

Prior observations with the krt4:lifeact and krt4:utrch (figure 3A) transgenic lines reveal that both keratinocyte layers will move distally after burn application.

- Pg. 7 - "The axons of sensory neurons are ensheathed within actin-rich channels running through basal keratinocytes 50,51." ref 51 is a *C. elegans* paper which does not have basal keratinocytes.

This was in error. The correct reference has replaced reference 51 (O’brien, *J Comp. Neurol.*, 2012), in which electron microscopy is used to document the development of two layers of epithelial cells that also ensheath sensory neurons in a protective manner similar to glial cells in the central nervous system.

- Figures S1E and F - the authors state that RB and DRG soma don't move. However, it was unclear from the figure panels and legend whether the authors imaged neurons that actually innervate the caudal fin (rather than some other region of the animal). Please clarify. For comparison, Fig S1F needs a pre-injury image to be meaningful.

The imaged cell bodies were those in the posterior trunk region, which are responsible for innervating the posterior sections of the fish including the caudal fin. From our observations, there was no movement of neuronal cell bodies after the burn.

- Figure 5 title - can the authors clarify what aspect of this figure relates to "sustained epidermal damage"

The figure 5 title has been updated in response to the reviewer comments.

- Figure 6 - is touch sensitivity really "restored" as the authors suggest? Alternatively, sensitivity may never be lost in isotonic treatment. Or the loss may be delayed?

We have modified the text accordingly by updating our phrasing – “restored” has been replaced with “improved” to indicate benefit over time.

- Can the authors further disentangle the effects of keratinocyte migration, ROS, and isotonic treatment on axon regeneration? For example, would the addition of CK666 to the Isotonic +1 hpw treatment improve axon regeneration? Can the authors directly manipulate ROS signaling (e.g., through exogenous addition of H2O2 or duox1 MO) to alter regeneration outcomes in their wounding assays?

See the comments above.

- Figure 6 title - consider removing or clarifying the word "excessive" here

The title has been revised according to the reviewer suggestion.

- hpw vs hpb were used inconsistently throughout the text

The manuscript has been revised to use “hpw” when referring to the timeframe after injury application.

Methods:- Zebrafish transgenics are missing allele namesReferences:- Many mistakes were noted in this section e.g., journal names missing, wrong authors, typos, DOIs misformatted

The references section has been corrected to use formatting consistent with APA citation and eLife preferred guidelines.